

# CODETASTE: Can LLMs Generate Human-Level Code Refactorings?

**Alex Thillen** [1 2]   **Niels Mündler** [1]   **Veselin Raychev** [2]   **Martin Vechev** [1]

## Abstract

LLM coding agents can generate working code, but their solutions often accumulate complexity, duplication, and architectural debt. Human developers address such issues through refactoring: behavior-preserving program transformations that improve structure and maintainability. We investigate whether agents (i) can execute refactorings reliably and (ii) identify the refactorings that human developers actually chose in real codebases. To this end, we construct CODETASTE, a benchmark mined from large multifile open-source refactorings. To score solutions, we combine repository test suites that measure functional correctness with tailored static checks that verify removal of undesired and introduction of desired code patterns using dataflow reasoning. Our results show a clear gap: agents perform well at implementing refactorings that are specified in detail, but often fail to discover the human refactoring choices when given a focus area for changes. A propose-then-implement decomposition improves alignment, and selecting the best-aligned proposal before implementation can yield further gains. CODETASTE provides an evaluation target and a potential preference signal for aligning coding agents with human refactoring decisions in realistic codebases. We release the benchmark, leaderboard, and code[1].

## 1. Introduction

Coding agents are being rapidly adopted across the software engineering industry (Mürtz & Müller, 2025; Sarkar, 2025), mostly fueled by their recent impressive performance on autonomous resolution of real-world software engineering tasks (Jimenez et al., 2024; Badertdinov et al.,

[1]Department of Computer Science, ETH Zurich [2]LogicStar.ai, Zurich, Switzerland. Correspondence to: Alex Thillen <athillen@ethz.ch>, Niels Mündler <niels.muendler@inf.ethz.ch>.

*Proceedings of the 43rd International Conference on Machine Learning*, Seoul, South Korea. PMLR 306, 2026. Copyright 2026 by the author(s).

2025). However, success on issue resolution does not imply that agents can sustain code quality across iterations. SlopCodeBench (Orlanski et al., 2026) evaluates iterative specification updates and measures verbosity and structural erosion, showing that coding-agent-generated code becomes longer and more structurally strained as requirements evolve. A recent effort to build a web browser, including a rendering engine, using autonomous coding agents (Lin, 2026) illustrates some practical consequences: despite making notable early progress, the agents effectively stalled, unable to continue extending a codebase they had themselves created. Similarly, an experiment tasking a team of agents to autonomously build a C compiler (Carlini, 2026) produced a functional 100,000-line codebase capable of compiling the Linux kernel, but resulted in rigid, inefficient code that lacked human-like abstraction (Lattner, 2026). These observations suggest that while current methods excel at producing functional patches, they struggle to produce code that remains comprehensible and extensible over time. Meanwhile, humans address these issues through refactoring, the software engineering practice of applying behavior-preserving transformations that improve structure and maintainability (Fowler, 2018).

**Lack of Challenging Refactoring Benchmarks**   While benchmarks for performance at code generation and repair (Chen et al., 2021; Jain et al., 2025; Jimenez et al., 2024; Badertdinov et al., 2025) have recently scaled with increasing model competence, benchmarks for refactoring capabilities are lagging behind. While OpenAI reports that they train and evaluate their models on large-scale code refactoring tasks, their evaluation framework remains undisclosed (OpenAI, 2025a). The wider community is therefore limited to available open benchmarks, which are limited in size and diversity: Across 6 recent benchmarks for code refactorings, most are limited to single-file edits, and all are restricted to a single programming language (Kovacic et al., 2025; Aider-AI, 2024; Cassano et al., 2023; Liu et al., 2025; Xu et al., 2026; Gautam et al., 2025).

This limitation causes these benchmarks to quickly become obsolete. For example, GPT-5.1 CODEX already achieves 75% accuracy on SWE-Refactor, a benchmark proposed

[1]Benchmark, leaderboard, dataset, and code are available at: https://codetaste.logicstar.ai/

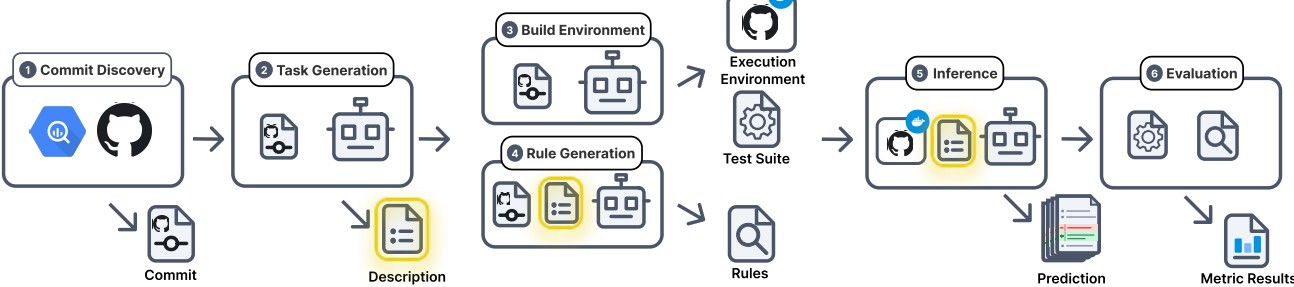

*Figure 1.* Overview of the CODETASTE benchmark pipeline. The workflow follows six key stages: (1) Commit Discovery from GitHub; (2) Generation of Task Descriptions; (3) Build Environment, which creates a reproducible execution environment for tests and agent inference; (4) Rule Generation of static analysis rules that capture refactoring intent; (5) Inference, where the agent generates a patch; and (6) Evaluation of generated patches against static analysis rules and test suites.

only 2 months prior to the model release. Beyond this size limitation, these benchmarks measure capabilities at following well-specified refactoring instruction, and do not measure whether coding agents can determine adequate refactorings autonomously. This distinction matters for the practical application of coding agents: to bootstrap a code base out of accumulated technical debt, agents must recognize when code has become unmaintainable and determine what transformations are necessary to restore clarity (Perry & Wolf, 1992). Meanwhile, existing evidence suggests that LLMs struggle at this task: A recent study found that LLMs rarely identify refactoring opportunities on their own, with LLMs discovering less than a fifth of human-proposed refactoring opportunities (Liu et al., 2025).

**Our Work** In this work, we investigate this challenge extensively and use precise code-level measurements to assess whether state-of-the-art coding agents can autonomously discover and implement relevant refactorings in real-world code bases. We introduce an alignment score that combines functional-correctness preservation with how closely a generated patch reflects the intended human refactoring. Our results indicate a clear gap: while state-of-the-art coding agents score highly on implementing complex refactorings that are specified in detail, achieving up to 70% alignment in our evaluation, they fail to mitigate undesired coding patterns correctly when no concrete refactoring is specified, achieving less than 8% alignment in the direct Open Track setting. We propose and release the benchmark designed for this task, CODETASTE, to the public, along with a leaderboard of recent coding agents, and our code implementation.

**Scalable Pipeline for Refactoring Instances.** We mine popular GitHub repositories for large multi-file refactorings performed by human developers. We then filter and transform these to 100 high-quality refactoring tasks. For each task, we generate an executable environment and

static analysis rules that match the changed code patterns. The latter support semantic dataflow reasoning within files. This provides more flexibility than syntactic matching while also supporting automatic verification. After the coding agent applies a refactoring, we measure performance, ensuring that (i) the repository test suite passes and (ii) the static analysis rules confirm code pattern transformation.

**Two Complementary Tracks for Evaluating Agent Performance.** Our benchmark features two tracks: In the *Instructed Track* the agent is tasked to correctly implement a detailed refactoring. Frontier models achieve alignment scores of up to 69.6%, indicating both correct implementations and consistent transformation. The *Open Track* evaluates whether agents can propose refactorings that align with human developer choices. In this track, the agent is only tasked to improve a general focus area. We discover that agents struggle significantly here: direct Open Track alignment remains below 8%, and even oracle-supported selection reaches only up to 20.6%. Our case studies show that current agents often take shortcuts and tend towards applying minimal, insufficient changes unless explicitly instructed to plan out a refactoring.

We observe that, still, agent rankings on our benchmark correlate with rankings on other coding benchmarks (Jimenez et al., 2024; Mündler et al., 2024), with for example GPT-5.2 ranking above QWEN3, with the former achieving 7.9% alignment in the Open Track direct setting while the latter scores only 2.0%. This suggests that human refactoring preferences are already partially captured by general model training.

## 2. Methodology

CODETASTE is constructed using a simple, six-step pipeline, summarized in Figure 1. Starting from a dataset of historic code changes, it (i) mines large, multi-file refactorings performed by human developers, (ii) translates each

*Figure 2.* Data funnel for obtaining the CODETASTE benchmark instances. In each step of the main commit pipeline, we filter for the top $n$ entries using (1) keywords, (2) complexity checks, (3) LLM assessment. From the resulting entries, we extract 100 instances for which our agentic setup produces (4) a working execution environment and (5) valid static analysis rules.

refactoring into task descriptions for our two tracks, (iii) builds a reproducible execution environment that can run each repository's unit tests, and (iv) generates static analysis rules that capture desired and undesired patterns of the transformation. The pipeline then (v) runs coding agents in the resulting environments and (vi) evaluates their patches with tests and static rules. We use this pipeline to generate 100 challenging refactoring tasks.

For evaluation, coding agents are run inside the execution environment and tasked to produce a candidate patch that implements the described refactoring. To allow agents to leverage their full capabilities, they can interact with the task description, the codebase, the test suite, and the execution environment, to produce a final predicted patch, which is simply extracted as the applied changes to the codebase. Finally, we evaluate this patch using (i) the test suite to validate whether the functionality has been preserved and (ii) the static analysis rules to validate how consistently the refactoring was implemented.

### 2.1. Notation and Definitions

We introduce the notation to describe test suites of codebases, changes to codebases in the form of patches, static analysis rules, and derived metrics.

**Test Suites** A codebase $R$ is a collection of code files. On any codebase $R$, a set of tests $T$ can be used to check the correctness of the code files. Executing a test suite $T$ of repository $R$ runs a total number of tests $|T|$ and returns $\exec_R(T) = (P_R, F_R)$ as counts of passing tests $P_R$ and failing tests $F_R$. If execution of the test suite fails unexpectedly, e.g., due to a crash, we set $(P_R, F_R) = (0, 0)$.

**Codebases and Patches** Following the notation of Mündler et al. (2024), patches $X$ are collections of line changes to files, and $R \circ X$ denotes codebase $R$ after applying patch $X$. In this case, the human-authored *base state* is $R$. The developer written *golden patch* is denoted $X^*$, and the corresponding *golden state* is $R \circ X^*$.

Let $L_R$ be the set of lines in $R$, where each line $l \in L_R$ is a triple $(f, n, s)$ representing its file, line number in the file, and line content, in this order. A patch $X$ induces a partial mapping $\mathcal{M}_X : L_R \rightharpoonup L_{R \circ X}$ tracking line persistence.

Two lines $l \in L_R$ and $l' \in L_{R \circ X}$ are *unchanged* in patch $X$ ($l \equiv_X l'$) iff $\mathcal{M}_X(l) = l'$. Lines deleted by $X$ are excluded from $\dom(\mathcal{M}_X)$, while new lines in $R \circ X$ lack a preimage. Edited lines are represented by a deleted line and a new line, respectively. We define the sets of *added* lines $L_X^+ = L_{R \circ X} \setminus \im(\mathcal{M}_X)$, the *removed* lines $L_X^- = L_R \setminus \dom(\mathcal{M}_X)$ and *changed* lines $L_X = L_X^+ \cup L_X^-$.

**Static Analysis Rules** A static analysis rule $\gamma$ is a declarative query over code, expressed as a pattern over abstract syntax tree nodes, with optional additional constraints on language, filename, or semantic properties like taint flow (Bull et al., 2002; Semgrep). For a codebase $R$, if the query pattern matches lines of code $L \subseteq L_R$, and the additional constraints are satisfied, the rule $\gamma$ *matches* the *witness* $\omega = L$. All matches of $\gamma$ on codebase $R$ form a multiset $\Omega(\gamma, R) = \{\omega_1, \ldots, \omega_k\}$.

We derive two metrics from $\Omega(\gamma, R)$:

1. *Match Count* $\mathcal{M}(\gamma, R) = |\Omega(\gamma, R)|$: the number of witnesses of a given rule $\gamma$ in codebase $R$, and

2. *Line Coverage* $\mathcal{C}(\gamma, R) = \bigcup_{\omega \in \Omega(\gamma, R)} \omega \subseteq L_R$: the subset of all lines in $R$ that are covered by $\gamma$.

Figure 3 illustrates these quantities for a pair of rules $(\gamma^-, \gamma^+)$ that capture the refactoring of a selector pattern in JavaScript. Each old selector call is a witness for $\gamma^-$, and each new store-hook call is a witness for $\gamma^+$. The match count of each rule is 1, while the total matches cover 4 lines of code in $\hat{X}_1$ and 2 lines of code in $\hat{X}_2$. Figure 3 further illustrates the intended level of abstraction: both changes satisfy the same additive rule $\gamma^+$ despite different hook names and formatting, because the rule encodes the structural pattern rather than a line-exact patch. A separate example with symbolic propagation is provided in Appendix B.

## 3. Benchmark Construction

In this section, we outline our benchmark construction pipeline, sketched in Figure 2.

### 3.1. Data Curation

Each task instance in our benchmark corresponds to a commit in a codebase. We obtain a list of commits for further processing using a three-stage pipeline that is summarized in the left half of Figure 2. The full details on the pipeline can be found in Appendix A.1.

We first query and keyword-filter candidate commits to prioritize 5000 commits that perform large refactorings. Second, we prefilter to around 1000 commits by removing repositories that are not publicly accessible anymore, commits that perform documentation-only changes, and code patches that are excessively large or too small to be relevant. Third, we assess commit messages and patches with a small LLM, tasking it to assign scores along simple categories, and keep the highest-ranked 160 candidates.

We run these three steps four times, once for each of the years 2023, 2024, and 2025 from the GitHub Archive (Grigorik, 2011), and once for the GitHub Activity Data dataset (GitHub, 2016), each time producing 160 instances. From these 640 instances, we choose the best 100 by score. This forms the set of instances for which we then generate tasks, execution environments and static rules.

### 3.2. Task Generation

For each task instance, we use an LLM to generate a highly detailed task description following a fixed format. The LLM is instructed to structure the task description as three components: 1) *Title:* A brief heading for the refactoring task. 2) *Summary:* A concise overview of the technical changes, and 3) *Why:* The rationale and relevance of the refactoring, with the goal of providing a highly detailed task description structured similarly to the instructions a human could write for a coding agent. As input, the LLM is provided with the commit message, associated Pull Requests, including titles, bodies, and linked issues, and the code of the golden patch $X^*$. The prompt used for this generation can be found in Figure 14.

### 3.3. Build Environment

To provide a stable and reproducible execution environment, our method utilizes a multi-phase bootstrap pipeline that pairs each benchmark instance with a specialized containerized environment.

**Phase 1: Agentic Environment Setup**  For each instance, our method starts with a containerized base environment into which it places a copy of the pre-refactoring codebase $R$ and the post-refactoring codebase $R \circ X^*$. We provide further details on the concretely used environments in Appendix A.2. Then, a coding agent is initialized inside the container and tasked with installing any additionally required system dependencies and creating a script $s$ that allows reproducing a standardized execution environment for the repository and orchestrates running its test suite. We provide the prompt in Figure 12 and an example excerpt of the generated scripts in Figure 13.

Upon completion of the environment setup by this agent, our pipeline validates the reproducibility of the environment by executing the pre- and post-refactoring test suites $\text{exec}_R(T)$ and $\text{exec}_{R \circ X^*}(T)$ using the standardized execution script $s$. A bootstrapping process is deemed successful only if at least $n$ test cases are discovered and at least $\alpha\%$ of all discovered test cases pass, or formally:

$$\forall R' \in \{R, R \circ X^*\}. \quad |T_{R'}| \geq n \quad \text{and} \quad \frac{P_{R'}}{|T_{R'}|} \geq \alpha$$

In our main construction run, we set $n = 10$ and $\alpha = 30\%$. During test evaluation we set a limit to the time and storage consumed by the test suite execution, limiting execution time to 15 minutes, and writable storage to 5GB.

**Phase 2: Runtime Creation**  After validation, the resulting image is transformed into a hardened runtime to prevent leaking the golden state and standardize the interface for agent evaluation. First, in order to ensure that the golden state $R \circ X^*$ is inaccessible from the base state $R$ during inference, the pipeline removes the repository's commit history and any network access from inside the container. Second, we add a unified entrypoint script that serves as the primary orchestration layer during benchmark inference. This script is executed upon container start to orchestrate the coding agent under test and extract the generated patch $\hat{X}$ for later evaluation.

### 3.4. Rule Generation

To capture the semantic intent of a code transformation without overfitting to specific concrete changes, we describe desired code changes using static analysis rules.

**Additive and Reductive Rules**  We distinguish two types of rules: additive and reductive rules. The additive rules $\Gamma^+$ describe desired code patterns, that should be introduced by a refactoring. The reductive rules $\Gamma^-$ describe code patterns that are undesired and should be removed.

**Rule Discovery**  We use a coding agent to describe the refactoring of the repository in terms of additive and reductive rules that match the post-refactoring and pre-refactoring states of the repository, respectively. For this task, the agent is provided with (i) the task description obtained in §3.2, (ii) read-only access to the pre-refactoring repository $R$ and (iii) specialized commands to register generated rules. This setup deliberately prevents the model

```
γ⁻
1  pattern: useStoreSelector($SELECTOR)
```

```
X̂₁
1  @@ -29,7 +29,9 @@ interface EventsChartProps extends BoxProps {
2   const EventsChart: ReactFCWithChildren<EventsChartProps> = ({
       events, ...rest }) => {
3 -   const { theme } = useStoreSelector(
        (state) => state.settings) // ω₁⁻
4 +   const theme = useSettingStore(
5 +     (state) => state.theme // ω₁⁺
6 +   )
7
8     const chartRef = useRef<any>(null)
```

```
γ⁺
1  pattern: $HOOK($STATE => $STATE.$FIELD)
2  metavariable-regex:
3    metavariable: $HOOK
4    regex: ^use[A-Z][a-zA-Z]*Store$
```

```
X̂₂
1  @@ -29,7 +30,7 @@ interface EventsChartProps extends BoxProps {
2   const EventsChart: ReactFCWithChildren<EventsChartProps> = ({
       events, ...rest }) => {
3 -   const { theme } = useStoreSelector(
        (state) => state.settings) // ω₁⁻
4 +   const theme = useSystemStore(state => state.theme) // ω₁⁺
5
6     const chartRef = useRef<any>(null)
```

*Figure 3.* Example additive rule $\boxed{\gamma^+}$ and reductive rule $\boxed{\gamma^-}$ from a migration away from a centralized selector helper toward domain-specific store hooks. Lower panels show two distinct candidate patches, $\hat{X}_1$ and $\hat{X}_2$, for the same initial repository state. Colored tokens show metavariable bindings, e.g., `$HOOK` denotes the bound store-hook name, while red/green line highlights show line coverage $\mathcal{C}$. In the shown excerpts, $\Omega(\gamma^-, R) = \{\omega_1^-\}$ and $\Omega(\gamma^+, R \circ \hat{X}_i) = \{\omega_1^+\}$ for $i \in \{1, 2\}$, so each displayed rule application has match count $\mathcal{M} = 1$; the additive witness in $\hat{X}_1$ spans several added lines.

from seeing the post-refactoring state $R \circ X^*$ or the refactoring patch $X^*$ itself, in order to avoid overfitting.

To ensure that the generated rules are correctly describing actual changes to the repository, the rule registration tool provided to the agent performs an additional step of validation, distinguishing additive and reductive rules.

The additive rules may not have any witnesses in the pre-refactoring state $R$. Conversely, to ensure the rule actually describes a desired pattern that was also implemented by a human developer, it has to have at least one witness in the golden post-refactoring state $R \circ X^*$.

$$\forall \gamma \in \Gamma^+ : \mathcal{M}(\gamma, R \circ X^*) > 0 \land \mathcal{M}(\gamma, R) = 0$$

Meanwhile, reductive patterns must not have witnesses in $R \circ X^*$ but must have at least one witness in $R$.

$$\forall \gamma \in \Gamma^- : \mathcal{M}(\gamma, R) > 0 \land \mathcal{M}(\gamma, R \circ X^*) = 0$$

To complete the rule generation step, the agent uses an additional tool call that signals completion. We detail the available tools in Appendix A.3.

**Additional Filtering** The previous steps can produce additive patterns that are overly strict: While they match actual human choices made during the refactoring, the choices might be arbitrary and not expected to be made similarly by other human developers. Such rules can concern for example exact naming of introduced classes or variables. To filter out such rules, we apply an additional heuristic filter on the generated rules. To this end, we prompt an LLM to classify whether a given rule enforces an arbitrary implementation choice, and filter out rules flagged

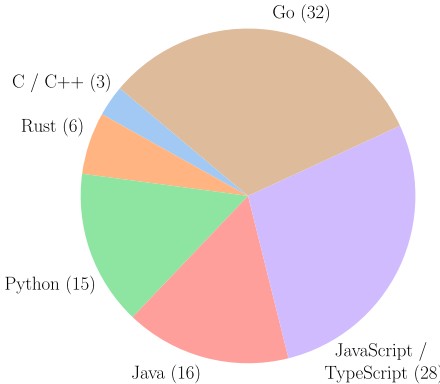

*Figure 4.* The 100 instances of CODETASTE span 6 different programming languages, focusing on JavaScript, TypeScript, and Go.

by the LLM. We validate the choices on a small sample by a human expert and find a high agreement of $80\%$. We provide additional details in Appendix A.3.

### 3.5. CODETASTE

We apply the described pipeline, using SONNET 4.5 as the main LLM to synthesize task descriptions, set up build environments and generate static rules.

**Dataset Overview** We thus obtain CODETASTE, our main benchmark, consisting of 100 instances distributed across 87 repositories and 6 programming languages. The detailed distribution across languages is shown in Figure 4, and Appendix A.7 reports mean alignment scores by primary language. Beyond the 100 instances evaluated in this paper, we release an additional 150 mined refactoring commits with task descriptions and static analysis rule sets to support future benchmark expansion. The benchmark comprises complex and large refactorings, as shown in Figure 5,

Figure 5. Summary statistics for CODETASTE instance metrics. Each instance is assessed by a significant number of rules and requires considerable file changes.

|  |  | Mean | Max |
|---|---|---|---|
| Task Prompts (# Characters) | Instructed Track | 1921.02 | 3451.0 |
|  | Open Track | 38.00 | 68.0 |
| Evaluation | Total Tests | 1638.53 | 12449 |
|  | Additive Rules | 17.94 | 70 |
|  | Reductive Rules | 63.41 | 120 |
|  | Total Rules | 81.35 | 160 |
| Golden Solutions | Files Changed | 91.52 | 290 |
|  | Lines Added | 1190.33 | 7172 |
|  | Lines Removed | 1415.06 | 11649 |
|  | Lines Edited | 2605.39 | 18821 |

with instances on average requiring edits to $91.52$ files and $2605.39$ lines of code. The most complex tasks require up to $18821$ line changes across up to $290$ files. To validate correctness, the benchmark runs an average of $1638.53$ tests and validates an average of $17.94$ additive and $63.41$ reductive rules per instance.

**Task Inference** The benchmark evaluates coding agents on the base commit $R$ within the pre-configured execution environment, where they are provided with a refactoring task. The agents are free to navigate and modify the entire environment, but have restricted access to the internet to prevent them from obtaining solutions from the original repository. Upon completion, we extract a generated patch $\hat{X}$ from the modifications to the repository state.

**Tracks** Regarding the refactoring task, the benchmark features two main tracks. In the *Instructed Track*, the agent is provided with a detailed description of the refactor to implement. This track mostly evaluates the coding agent's capability to perform long-context tasks consistently. These tasks contain significant detail, as shown in Figure 5, with on average $1921.02$ characters per description.

The goal of the *Open Track* is to vaguely define the area of the desired refactoring, or a problematic aspect of the code base, to avoid completely random exploration, but not to reveal directly what is desired as a refactoring. We derive the Open Track description via a secondary LLM pass over the Instructed Track full description. In the end, we manually investigated all Open Track descriptions to confirm that they indeed do not reveal the actual refactoring. Examples of both description types are shown in Figures 14 and 15. The average description is significantly shorter than in the Instructed Track, with only $38$ characters on average.

**Open Track Modes** To allow for fine-grained understanding of the refactoring agent decisions in the Open Track, we define three different modes of inference sketched in Figure 6. In *Direct* mode, the agent implements

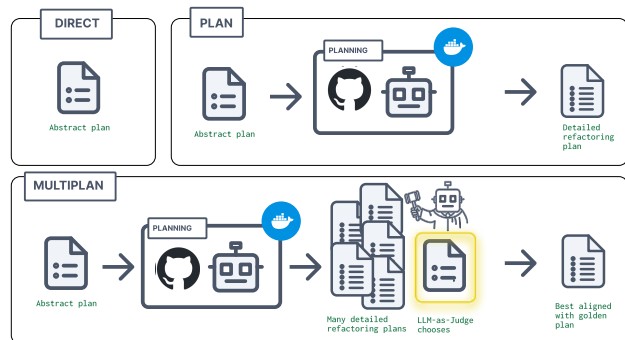

Figure 6. The Open Track of CODETASTE has three distinct inference modes: *Direct*, in which the agent directly performs the refactoring, *Plan*, in which the agent drafts a plan before execution, and *Multiplan*, in which an oracle picks the best-aligned self-drafted plan of the model before execution.

changes directly from the instruction. In *Plan* mode, the agent first generates a single implementation plan. Then, the agent is reinitialized and prompted to execute this plan in a second pass. In *Oracle Multiplan* mode, the agent generates multiple candidate plans. An LLM-based oracle judge with access to the full refactoring description then selects the plan that is closest to the refactoring. Finally, the agent is reinitialized to execute the selected plan. The used prompts are provided in Figures 16–18.

## 4. Experimental Evaluation

In this section, we report the results of our extensive evaluation of coding agents on CODETASTE.

### 4.1. Experimental Setup

**Agent Configurations** We evaluate five coding agents across task specifications. Our evaluated configurations include GPT-5.2 and GPT 5.1 Codex Mini (GPT-5.1 M, Singh et al., 2025), Claude 4.5 Sonnet (SONNET 4.5, Anthropic, 2025), the 229B-parameter MiniMax M2.7 (M2.7, MiniMax, 2026), and the instruct version of the open-source Qwen3 Coder 30B A3B (QWEN3, Yang et al., 2025). We use model-developer-recommended agent harnesses where applicable: Codex CLI (OpenAI, 2025b) for the OpenAI models, Claude Code (Code Claude Docs, 2025) for SONNET 4.5 and MINIMAX M2.7, and Qwen Code (QwenLM, 2025) for QWEN3. The Codex CLI is explicitly configured with reasoning effort `medium`, while Claude Code and Qwen Code use their respective defaults. We use SONNET 4.5 as the judge for plan selection in Oracle Multiplan mode.

**Functional Correctness** To evaluate functional correctness of a predicted patch $\hat{X}$, we require that the test suite does not report a major regression. Concretely, we require a minimum number $p_{\min}$ of test cases to pass on the patched

code base $R \circ \hat{X}$, and a maximum number $f_{\max}$ of test cases to fail. To determine $p_{\min}$ and $f_{\max}$, we execute the test suites of both $R$ and $R \circ X^*$ for $k$ times. We then set $p_{\min} = \min_{i \in \{1, \ldots, k\}}(P_R^i, P_{R \circ X^*}^i)$ and $f_{\max} = \max_{i \in \{1, \ldots, k\}}(F_R^i, F_{R \circ X^*}^i)$, where $P_R^i$ denotes the number of passing tests of the $i$-th run on codebase $R$, and $F_R^i$ failing tests respectively. Finally, we define functional correctness as $\text{PASS}(\hat{X}) = \mathbb{1}\left[F_{R \circ \hat{X}} \leq f_{\max} \land P_{R \circ \hat{X}} \geq p_{\min}\right]$. We set $k = 5$ to account for flaky tests, following prior work (Mündler et al., 2024).

**Instruction Following Rate**   We assess adherence to the refactoring intent using the discovered static analysis rules $\Gamma$. Let $\mathbb{1}_{\mathcal{M}}(\gamma, R) = \mathbb{1}[\mathcal{M}(\gamma, R) > 0]$ indicate that rule $\gamma$ matches in state $R$. The instruction following rate (IFR) measures the recall of additive and reductive rules in the resulting state $R \circ \hat{X}$. The goal is to reward patches that produce code matching additive rules and penalize patches that fail to remove reductive rules. Consequently, the additive score IFR$^+$ is the percentage of rules $\gamma \in \Gamma^+$ that match on the predicted state of the repository $R \circ \hat{X}$, and the reductive score IFR$^-$ is the percentage of rules $\gamma \in \Gamma^-$ that no longer match there. The total instruction following score IFR is a weighted combination of the scores. By construction, $\text{IFR}(X^*) = 1$.

$$\text{IFR}^+(\hat{X}) = \frac{1}{|\Gamma^+|} \sum_{\gamma \in \Gamma^+} \mathbb{1}_{\mathcal{M}}(\gamma, R \circ \hat{X})$$

$$\text{IFR}^-(\hat{X}) = \frac{1}{|\Gamma^-|} \sum_{\gamma \in \Gamma^-} \left(1 - \mathbb{1}_{\mathcal{M}}(\gamma, R \circ \hat{X})\right)$$

$$\text{IFR}(\hat{X}) = \frac{|\Gamma^-|}{|\Gamma|} \text{IFR}^-(\hat{X}) + \frac{|\Gamma^+|}{|\Gamma|} \text{IFR}^+(\hat{X})$$

**Alignment Score**   We combine the instruction following rate and the functional correctness into an *alignment score* $\mathcal{A}$. Concretely, the alignment score $\mathcal{A}(\hat{X}) = \text{PASS}(\hat{X}) \times \text{IFR}(\hat{X})$. The multiplication ensures that adherence to refactoring intent is only rewarded if the generated patch is also functionally correct.

**Precision**   Precision $\text{PREC}(\hat{X})$ measures how well $\hat{X}$ avoids unrelated changes. For reported agent diff-size and precision metrics, we compute $L_{\hat{X}}^+$ and $L_{\hat{X}}^-$ after filtering

out build, vendor, documentation, configuration, comment-only, and whitespace changes. For a codebase $R$, $\mathcal{C}_\Gamma(R) = \cup_{\gamma \in \Gamma} \mathcal{C}(\gamma, R)$ is the set of lines matched by the rule set $\Gamma$. We define additive precision PREC$^+$ as the percentage of added lines $L_{\hat{X}}^+$ that also match the additive rules $\Gamma^+$. Negative precision PREC$^-$ is the percentage of removed lines $L_{\hat{X}}^-$ that were matched by the reductive rules $\Gamma^-$ in $R$. Precision PREC is the weighted average of these terms.

$$\text{PREC}^+(\hat{X}) = \frac{|L_{\hat{X}}^+ \cap \mathcal{C}_{\Gamma^+}(R \circ \hat{X})|}{|L_{\hat{X}}^+|}$$

$$\text{PREC}^-(\hat{X}) = \frac{|L_{\hat{X}}^- \cap \mathcal{C}_{\Gamma^-}(R)|}{|L_{\hat{X}}^-|}$$

$$\text{PREC}(\hat{X}) = \frac{|L_{\hat{X}}^+|}{|L_{\hat{X}}|} \text{PREC}^+(\hat{X}) + \frac{|L_{\hat{X}}^-|}{|L_{\hat{X}}|} \text{PREC}^-(\hat{X})$$

### 4.2. Results on the Instructed Track

Even in the Instructed Track, where agents are provided with a detailed blueprint specifying the exact refactoring transformations required, notable performance gaps emerge among frontier systems. Unless otherwise specified, metrics in this section and §4.3 are instance averages.

**SOTA Agents Have Major Differences**   GPT-5.2 statistically significantly outperforms all other agents on $\mathcal{A}$ (Appendix A.8), achieving an average $\mathcal{A}$ of 69.6%. In contrast, GPT-5.1 M, SONNET 4.5, and MINIMAX M2.7 achieve only 34.4%, 32.1%, and 30.5%, respectively. The open-source QWEN3 lags substantially behind at 10.7%.

Notably, all of GPT-5.2, GPT-5.1 M and SONNET 4.5 have high IFR scores of 89.0%, 70.4% and 67.0%. However, the $\mathcal{A}$ of the latter two is significantly lower than for GPT-5.2, since their PASS is roughly 30 percentage-points below GPT-5.2's 76.0% at 47.0% and 43.0%, respectively. Thus, while all frontier agents are able to follow detailed and large-scale changes, there are large gaps in maintaining code correctness. In our case studies, we find that GPT-5.2 achieves higher PASS rates by applying targeted, individual patches, whereas SONNET 4.5 runs indiscriminate search-and-replace commands. While one might suspect that propensity to validate outputs with the test suite might have a strong effect, GPT-5.2 actually runs tests less frequently than SONNET 4.5.

**Overall High Precision**   Overall, all agents achieve similar precision scores, with GPT-5.1 M, SONNET 4.5, and GPT-5.2 achieving 50.2%, 50.1%, and 48.4%, respectively. These scores effectively cluster around human precision at 48.5%. This suggests that with clearly specified refactoring intent, frontier agents can keep changes in scope without substantial unrelated modifications.

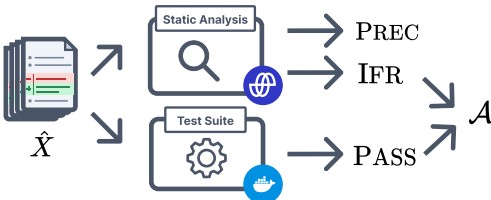

*Figure 7.* Sketch of the evaluation metrics for CODETASTE.

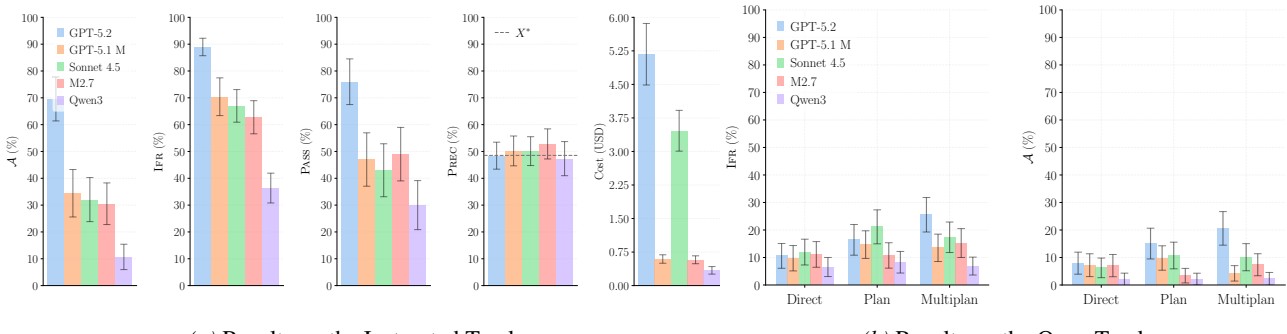

*(a)* Results on the Instructed Track          *(b)* Results on the Open Track

*Figure 8.* (Left) On the Instructed Track, GPT-5.2 significantly outperforms other models across IFR, PASS, and $\mathcal{A}$. In terms of $\mathcal{A}$ and PASS, QWEN3 significantly underperforms. All models achieve roughly human-level PREC. (Right) In the Open Track, the propose-then-implement strategy (Plan) and oracle selection (Oracle Multiplan) improve IFR and $\mathcal{A}$ for GPT-5.2. Functional correctness PASS is rarely impaired, resulting in similar $\mathcal{A}$ as IFR. Across all agents, performance is far from human refactoring choices.

**Thorough Refactorings Are Expensive** The higher alignment of GPT-5.2 is resource-intensive. To manage execution costs, agents have a $11 budget per task; over-budget runs are terminated and evaluated with available output. GPT-5.2 hit this limit in 14 instances, compared to only 3 instances for SONNET 4.5. The mean cost for GPT-5.2 in instructed mode is $5.17, compared to $3.46 for SONNET 4.5 and $0.59 for GPT-5.1 M. This reflects GPT-5.2's targeted patches, while SONNET 4.5 relies on cheaper search and replace commands (e.g. `sed -i`).

### 4.3. Results on the Open Track

Achieving high IFR on the Open Track is significantly harder. This is because agents are only provided with a high-level focus area and need to recover human architectural choices. In Direct Inference mode, the maximal achieved alignment score $\mathcal{A}$ over all agents is 7.9%, as can be seen in Figure 8b.

**Planning Helps Large Refactorings** When agents first propose a plan before implementing it (Plan Mode), there is a substantial increase in $\mathcal{A}$. For GPT-5.2, the $\mathcal{A}$ nearly doubles from 7.9% to 15.1%. On average across all agents, the scores increase by about 2.2 percentage points, corresponding to a 36% relative increase. This suggests that explicit reasoning before editing helps recover large-scale refactoring intents. Case studies (Appendix C.1) show that planning yields repository-wide transformations reflecting high-level architectural thinking. For GPT-5.2, GPT-5.1 M, and SONNET 4.5, the corresponding patches are approximately 2 to 3 times larger (measured by $|L_{\hat{X}}|$) and achieve up to a 76% relative increase in IFR. The broader scope can come at the cost of up to 23 percentage-points lower PASS rates (Direct vs. Plan). QWEN3 exhibits the largest decline, whereas GPT-5.2 sees no decline. Details across all agents and modes are presented in Table 5.

**Suggestions Generally Include Human Choices** Oracle Multiplan tests whether generated proposals include one closer to the human refactoring. Here, agents generate several proposals, and an LLM with oracle knowledge chooses the plan closest to the desired refactoring. GPT-5.2 reaches an $\mathcal{A}$ of 20.6%, statistically significantly above all other agents (Appendix A.8). Interestingly, SONNET 4.5 does not improve in this setting, with its score dropping slightly from 10.7% in Plan mode to 10.1%. This might indicate that, when tasked to generate several possible refactorings, the model shifts into a different, less aligned distribution of refactoring plans. A randomized six-plan control finds no evidence that the Oracle Judge biases plan selection (Appendix A.6). Across Open Track settings, we can see in Table 4 that IFR$^-$ is generally higher than IFR$^+$, suggesting that identifying and mitigating undesirable patterns is easier than reproducing the human-introduced design choices.

### 4.4. Failure Modes in Open Track

To answer what the Open Track actually tests, we investigated a sample of patches from GPT-5.2 and SONNET 4.5 and compared them with the golden patch. Some observed issues concern (a) fixating on salient subjects, e.g., a typo in a class name, ignoring the broader focus area, (b) resorting to lazy, hacky workarounds, e.g., creating a compatibility shim to bypass updating hundreds of import statements, (c) relying on destructive, repository-wide replace-all string commands, leaving the codebase in a broken and inconsistent state, and (d) working on entirely different aspects of the system than required. While we cannot predict what a realistic upper bound for human alignment is on a refactoring, we observe that even the best current agents have major issues choosing appropriate refactorings. We include examples of LLM-assisted manual analysis on such cases in Appendix C.

*Table 1.* Comparison of related benchmarks by scope and evaluation. Intent Alignment denotes whether the benchmark evaluates how well a generated patch recovers a human refactoring intent that is not provided to the agent. Tests denote execution-based behavior-preservation checks. Static checks denote task-specific structural or refactoring oracles, excluding generic metrics, parse checks, and coverage diagnostics. [*]Statistics computed by us. [1]Data is unavailable. [2]Lacks reference solution $X^*$.

| Benchmark | # Languages | Mean $|F_{X^*}|$ | Mean $|L_{X^*}|$ | Intent Alignment | Tests | Static Checks |
|---|---|---|---|---|---|---|
| MiniCode [20] | 1 | —[1] | —[1] | ✗ | ✓ | ✗ |
| Aider refactor benchmark [1] | 1 | 1.0 | —[2] | ✗ | ✗ | ✓ |
| Can It Edit? [6] | 1 | 1.0 | 13.8[*] | ✗ | ✓ | ✗ |
| RefactorMirror [25] | 1 | 1.0 | 25.3[*] | ✓ | ✓ | ✗ |
| SWE-Refactor [40] | 1 | 1.1[*] | 52.2[*] | ✗ | ✓ | ✓ |
| SWE-bench [19] | 1 | 1.7 | 32.8 | ✗ | ✓ | ✗ |
| RefactorBench [14] | 1 | 4.3 | 68.3[*] | ✗ | ✗ | ✓ |
| **CodeTaste (this work)** | **6** | **91.5** | **2605.4** | ✓ | ✓ | ✓ |

## 5. Related Work

As summarized in Table 1, no prior benchmark targets large, multi-file refactorings with intent-alignment evaluation across multiple languages.

**Single-File Refactoring** A number of benchmarks measure changes only on single-file and single-language edits. The Aider team(Aider-AI, 2024) and Can It Edit? (Cassano et al., 2023) benchmarked function-level refactoring in Python. RefactorMirror (Liu et al., 2025) and Cordeiro et al. (2024) evaluate single-file Java refactoring using human expert assessment and generic code complexity metrics. Rejiba et al. (2025) measure code complexity metrics on refactored C++ and C files.

**Multi-File Refactoring** MiniCode (Kovacic et al., 2025) evaluates extracting shared Python code into a library. SWE-Refactor (Xu et al., 2026) is a Java-only benchmark for few-file refactorings, with on average only 1.1 files. RefactorBench (Gautam et al., 2025) analyzes GPT-4o refactorings across on average 4.3 files on a Python dataset using hand-written source-code tests. CODETASTE spans six languages and requires significantly larger edits than prior work, making it a substantially more challenging benchmark, even in the Instructed Track.

**Repository-level Benchmarks** Spearheaded by Jimenez et al. (2024), evaluating coding agents on the autonomous resolution of real-world repository-level tasks was established as the gold standard for assessing LLM capabilities. While Jimenez et al. (2024) focuses on issue resolution, follow-up work proposed benchmarks on feature addition (Li et al., 2025; Du et al., 2025), unit test generation (Mündler et al., 2024), function generation (Liang et al., 2025), code performance (He et al., 2025), and security (Chen et al., 2026), but not on refactoring.

**Methods to improve refactoring** Pomian et al. (2024) evaluate whether models can decide when to extract code into external methods. Zhang et al. (2024) evaluate the function-level capability of models at rewriting Python code into idiomatic expressions. Shirafuji et al. (2023) similarly prompt GPT-3.5 with few-shot prompts to evaluate its refactoring capabilities. All of these methods are limited to very specific refactorings, while our benchmark covers a wide range of potential refactorings.

## 6. Limitations

CODETASTE contains 100 instances, so we treat small gaps between agents as suggestive rather than decisive, particularly in the Open Track. This scale partly reflects the cost of constructing the execution environment and rule set for each instance, as well as the cost of running frontier coding agents, where a single agent run on one task often exceeds USD $10. Further, the distribution across the six languages is uneven, and task difficulty may vary, limiting language-specific conclusions. Finally, Open Track alignment measures agreement with the original human refactoring, which does not capture all valid refactoring decisions. Although we audit additive rules that encode arbitrary implementation choices, CODETASTE relies on LLM-assisted construction steps, which may introduce systematic biases.

## 7. Conclusion

We introduced CODETASTE, a benchmark for evaluating refactoring agents on real, large, many-file changes mined from open-source repositories. Our results demonstrate that when provided with explicit refactoring instructions, current agents achieve relatively strong, yet unsaturated, performance. However, alignment collapses when only a focus area is provided. Propose-then-implement strategies and oracle selection over multiple plans increase alignment by up to 2.61-fold, yet remain far from human choices. These findings highlight that autonomous judgment about *what* to refactor remains a key bottleneck. We hope CODETASTE serves as a rigorous evaluation target and a preference signal for agents to preserve code quality long-term.

## Acknowledgements

This work was conducted as part of the grant SAFEAI (Certified Safe, Fair and Robust Artificial Intelligence). The work has received funding from the Swiss State Secretariat for Education, Research and Innovation (SERI), contract no. MB22.00088.

## Impact Statement

This paper presents work whose goal is to advance the field of Machine Learning. There are many potential societal consequences of our work, none of which we feel must be specifically highlighted here.

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

# A. Experimental Details and Ablations

In this section, we provide additional details about the implementation, hyperparameters, datasets, and so on.

## A.1. Data Curation Pipeline Details

We obtain a list of commits for further processing using a three-stage pipeline.

**Stage 1: Querying and Keyword Filtering**  We filter out all repositories with less than 90 stars. Among those, we search for commits that edit at least 45 files and feature specific keywords. Concretely, let $K^+$ and $K^-$ denote the sets of keywords indicative of refactoring and non-refactoring commits, respectively. We further distinguish between strong ($K^{\pm}_{\text{strong}}$) and weak ($K^{\pm}_{\text{weak}}$) keywords, which form subsets of $K^+$ and $K^-$. For instance, `deduplicate` $\in K^+_{\text{strong}}$, `reorganize` $\in K^+_{\text{weak}}$, `fix` $\in K^-_{\text{strong}}$, and `lint` $\in K^-_{\text{strong}}$.

Commit messages $M$ of candidate commits must satisfy:

$$\text{Count}(M, K^-_{\text{strong}}) = 0 \quad \text{and} \quad \frac{\text{Count}(M, K^+)}{\text{Count}(M, K^-_{\text{weak}})} \geq 2$$

We then obtain the top 5000 commits sorted lexicographically descending by $(\text{Count}(M, K^+_{\text{strong}}), \text{Count}(M, K^+_{\text{weak}}), \#\text{stars}, \#\text{files changed})$.

**Stage 2: Prefiltering**  We remove all repositories and commits that are not accessible anymore, i.e., have been deleted or made private. To exclude pure documentation changes, we require a minimum of 25 edits in files that have a file extension of typical programming languages, such as `.py` or `.cpp`. Via manual inspection, we determine that we need to remove extreme outliers, which include, for example, entire virtual environments. We retain only those commits that satisfy:

$$0.5p_{10} \leq \text{changes} \leq 2p_{90}$$

where changes refers to the number of added or removed lines, and the number of created, deleted, modified, or moved files. Finally, to exclude highly repetitive and simple search-and-replace refactorings, we sort the results by the ratio of $\frac{|\text{diff}_c|}{|\text{diff}|}$, where $|\text{diff}_c|$ is the size of the Gzip level 6 compressed diff (excluding file context) in bytes.

**Stage 3: LLM Scoring**  In the final step we use a small LLM (CLAUDE 4.5 HAIKU) to analyze the commit messages and code diffs. The prompts for this step can be found in Figures 10 and 11.

First, the LLM is prompted to score the commit message across several dimensions: a score for being a project-wide change $S_{\text{large}}$, a score for behavior-preserving refactoring $S_{\text{refactor}}$, and a message cohesion score $S_{\text{cohesion}}$. The model also estimates a length penalty $P_{\text{length}}$ for multi-topic messages, a triviality penalty $P_{\text{trivial}}$ for simple edits, and a penalty for general uncertainty $P_{\text{unc}}$. We retain the top 1,120 candidates based on the overall message score $S_{\text{msg}}$ calculated as:

$$S_{\text{msg}} = (0.35S_{\text{large}} + 0.45S_{\text{refactor}} + 0.2S_{\text{cohesion}}) \\ - (0.2P_{\text{length}} + 0.2P_{\text{trivial}} + 0.2P_{\text{unc}})$$

Next, the LLM is prompted to analyse the code diff, focusing on the refactoring likelihood $S_{\text{refactor}}$ and the estimated complexity $S_{\text{complexity}}$ of the change. It also identifies penalties for mixed content $P_{\text{mixed}}$, evidence of behavior-altering logic $P_{\text{risky}}$, signs of automation $P_{\text{auto}}$, uncertainty $P_{\text{unc}}$, and the presence of trivial changes $P_{\text{trivial}}$. The diff-specific score $S_{\text{diff}}$ is defined by:

$$S_{\text{diff}} = (0.8S_{\text{refactor}} + 0.2S_{\text{complexity}}) - 1.5(0.4P_{\text{mixed}} \\ + 0.2P_{\text{risky}} + 0.1P_{\text{unc}} + 0.15P_{\text{trivial}} + 0.15P_{\text{auto}})$$

We retain the top 160 candidates based on a combined score $S_{\text{total}}$ that prioritizes the diff analysis over the commit message:

$$S_{\text{total}} = \frac{S_{\text{msg}} + 3S_{\text{diff}}}{4}$$

**Running the Pipeline**  This pipeline is run four times, once each on the main table of the GitHub Activity Data dataset (GitHub, 2016) and the three main tables for 2023, 2024 and 2025 respectively in the GitHub Archive (Grigorik, 2011), filtered to PullRequestEvent for merged pull requests. We obtain the count of "stars", an indication of popularity, for each repository from donbarbos (2026). The four runs produce $4 \times 160 = 640$ candidates. For each candidate, we first assess the quality of the generated static rules and then confirm the correctness of the generated execution environment. We keep the first 100 candidates that pass both checks capping the downstream evaluation cost.

### A.2. Details of the Execution Environment Agent

We use a polyglot container based on Ubuntu 24.04 with pre-configured toolchains and version managers for common language ecosystems. The base image includes: (i) Python 3.8–3.11 managed via uv, (ii) Node.js 22.12.0 (default) managed via Node Version Manager (NVM), (iii) Go v1.23.4 and a Rust toolchain (cargo), (iv) OpenJDK, and (v) C/C++ via GCC/Clang.

The build environment agent, SONNET 4.5 in the CLAUDE CODE harness, is restricted to \$5 cost and 90 minutes time. To ensure portability of the benchmark, we limit the size of the writable storage layer, after all setup is completed, to 5GB. Further, we enforce a 15-minute time limit on running the test suite. This container is then committed.

In the runtime setup, we concretely remove the git commit history and remote origins to prevent leakage of golden commits.

### A.3. Details of the Rule Agent

In this section, we detail the specific commands that the rule discovery agent can use.

- BashExplore$_R$(cmd): Executes a set of read-only shell commands in the base codebase $R$ to identify relevant code patterns (e.g., ls, rg).

- Register($\Gamma_{\text{candidates}}$): Submits candidate rules in yaml syntax. Without loss of generality, we show the registration of additive rules only.

  **Registration:** Every valid additive rule is registered:

$$\Gamma^+ = \Gamma^+ \cup \{\gamma \in \Gamma_{\text{candidates}} \mid \mathcal{M}(\gamma, R \circ X^*) > 0 \\ \wedge \mathcal{M}(\gamma, R) = 0 \quad\quad\}$$

  To prevent redundant rules, candidates must be sufficiently distinct from previously registered rules; we require pairwise Jaccard overlap of matched lines to be at most 0.95.

  **Feedback:** For each $\gamma \in \Gamma_{\text{candidates}}$, the agent receives the match counts $\mathcal{M}(\gamma, R)$ and $\mathcal{M}(\gamma, R \circ X^*)$, as well as rejection reasons (if applicable). In the case of execution failures (e.g., SyntaxError or OpengrepError), the agent is forwarded parts of error message for refining the rules in the next iteration. For highly discriminative but invalid rules where $\mathcal{M}(\gamma, R \circ X^*) > 10 \times \mathcal{M}(\gamma, R)$, the agent is provided with a witness $\omega \in \Omega(\gamma, R)$ to facilitate adding exemptions.[2]

- Finish(): Signals completion and returns the valid registered rule set.

**Restrictive Additive Rule Audit**  For the final rule curation, we used Gemini 3.1 Pro to identify additive rules that enforce arbitrary implementation choices and filtered rules based on these decisions. To assess this filter, we audited 30 randomly selected additive rules with a human expert; Table 2 reports the human-Gemini agreement, including Cohen's $\kappa$.

---

[2]In many real-world refactorings there are edge cases such as unadjusted examples that can prevent a rule from reaching $\mathcal{M}(\gamma, R) = 0$. Providing witnesses for these matches allows the agent to construct more general patterns by exempting specific outliers.

*Table 2.* Human audit of restrictive additive rules.

| Human label | Gemini keeps rule | Gemini filters rule |
|---|---|---|
| Non-restrictive | 14 | 1 |
| Restrictive | 5 | 10 |
| Agreement | 24/30 (80%) | |
| Cohen's $\kappa$ | 0.60 | |

## A.4. Complete List of Used Repositories

We provide a complete list of the 87 repositories included in CODETASTE in Table 3. We have checked the license for all repositories and confirm they are compliant to be used in CODETASTE.

*Table 3.* Complete list of used repositories.

| | | |
|---|---|---|
| 99designs/gqlgen | Automattic/wp-calypso | DataDog/datadog-agent |
| NationalSecurityAgency/ghidra | PrestaShop/PrestaShop | TanStack/router |
| actualbudget/actual | ansible/ansible | antvis/G6 |
| apache/flink | apache/hadoop | apache/nuttx |
| apache/pinot | apache/pulsar | apache/shardingsphere |
| apache/shenyu | aquasecurity/tracee | aws/aws-cli |
| badges/shields | bentoml/BentoML | bevyengine/bevy |
| boa-dev/boa | burn-rs/burn | bytebase/bytebase |
| chaos-mesh/chaos-mesh | chroma-core/chroma | cilium/cilium |
| clientIO/joint | conan-io/conan | containerd/nerdctl |
| cosmos/cosmos-sdk | deepset-ai/haystack | elastic/eui |
| etcd-io/etcd | ethereum/go-ethereum | excalidraw/excalidraw |
| fabricjs/fabric.js | getsentry/sentry | ggerganov/llama.cpp |
| go-gitea/gitea | gohugoio/hugo | golang/go |
| golangci/golangci-lint | google/gvisor | google/tink |
| gopasspw/gopass | gradle/gradle | graphhopper/graphhopper |
| hashicorp/consul | highcharts/highcharts | hrydgard/ppsspp |
| huggingface/transformers | immich-app/immich | influxdata/influxdb |
| istio/istio | jerryjliu/llama_index | juju/juju |
| kedro-org/kedro | knative/serving | koajs/koa |
| kubevirt/kubevirt | lima-vm/lima | loft-sh/vcluster |
| mne-tools/mne-python | mockito/mockito | mongodb/mongo-go-driver |
| monkeytypegame/monkeytype | n8n-io/n8n | netty/netty |
| nhost/nhost | opensearch-project/OpenSearch | prisma/prisma |
| pulumi/pulumi | pyvista/pyvista | remarkablemark/html-react-parser |
| ruffle-rs/ruffle | rust-lang/rust | saleor/saleor |
| sanity-io/sanity | scikit-learn/scikit-learn | syncthing/syncthing |
| temporalio/temporal | tinymce/tinymce | twosigma/beakerx |
| vitessio/vitess | wader/fq | wikimedia/apps-android-wikipedia |

## A.5. Complete Data for Instruction Following and Test Passing

We provide detailed results on the instruction following, alignment score, test pass rate, precision, and edited lines of code for the different models and settings in Tables 4–6. For completeness, we also report $\mathcal{A}^+$ and $\mathcal{A}^-$, which are defined with respect to only the IFR$^+$ and IFR$^-$, respectively, below.

$$\mathcal{A}^+(\hat{X}) = \text{PASS}(\hat{X}) \times \text{IFR}^+(\hat{X}) \quad \mathcal{A}^-(\hat{X}) = \text{PASS}(\hat{X}) \times \text{IFR}^-(\hat{X})$$

*Table 4.* Full experimental results across all tracks and models in percents on $\text{IFR}^+$ (additive instruction-following rate), $\text{IFR}^-$ (reductive instruction-following rate), and PASS (test pass rate).

| | | GPT-5.2 | | | GPT-5.1 M | | | SONNET 4.5 | | | MINIMAX M2.7 | | | QWEN3 | | |
| --- | --- | --- | --- | --- | --- | --- | --- | --- | --- | --- | --- | --- | --- | --- | --- | --- |
| | | $\text{IFR}^+$ | $\text{IFR}^-$ | PASS | $\text{IFR}^+$ | $\text{IFR}^-$ | PASS | $\text{IFR}^+$ | $\text{IFR}^-$ | PASS | $\text{IFR}^+$ | $\text{IFR}^-$ | PASS | $\text{IFR}^+$ | $\text{IFR}^-$ | PASS |
| Instructed Track | | 89.6 | 88.7 | 76.0 | 75.6 | 69.0 | 47.0 | 79.0 | 62.6 | 43.0 | 75.2 | 58.9 | 49.0 | 56.6 | 30.6 | 30.0 |
| Open Track | Direct | 9.7 | 10.3 | 87.0 | 7.5 | 9.7 | 79.0 | 10.8 | 12.1 | 66.0 | 7.3 | 11.5 | 71.0 | 4.4 | 6.7 | 58.0 |
| Open Track | Plan | 11.2 | 17.2 | 87.0 | 9.8 | 15.1 | 66.0 | 16.8 | 21.7 | 57.0 | 10.5 | 10.8 | 63.0 | 4.1 | 8.8 | 35.0 |
| Open Track | Multiplan | 17.8 | 26.7 | 81.0 | 9.7 | 14.2 | 58.0 | 17.4 | 18.0 | 50.0 | 10.4 | 15.9 | 58.0 | 5.3 | 7.0 | 43.0 |

*Table 5.* Full experimental results across all tracks and models for PASS, $\mathcal{A}$, IFR in percents with 95 % confidence intervals.

| | | PASS (%) | $\mathcal{A}$ (%) | $\mathcal{A}^+$ (%) | $\mathcal{A}^-$ (%) | IFR (%) | $\text{IFR}^+$ (%) | $\text{IFR}^-$ (%) |
| --- | --- | --- | --- | --- | --- | --- | --- | --- |
| **GPT-5.2** | | | | | | | | |
| Instructed | | $76.0_{\pm 8.5}$ | $69.6_{\pm 8.2}$ | $70.7_{\pm 8.6}$ | $69.6_{\pm 8.4}$ | $89.0_{\pm 3.3}$ | $89.6_{\pm 4.0}$ | $88.7_{\pm 4.0}$ |
| Open | Direct | $87.0_{\pm 6.7}$ | $7.9_{\pm 4.0}$ | $6.1_{\pm 3.2}$ | $7.9_{\pm 4.2}$ | $10.6_{\pm 4.5}$ | $9.7_{\pm 4.3}$ | $10.3_{\pm 4.7}$ |
| Open | Plan | $87.0_{\pm 6.7}$ | $15.1_{\pm 5.6}$ | $8.9_{\pm 4.5}$ | $16.1_{\pm 6.0}$ | $16.4_{\pm 5.6}$ | $11.2_{\pm 5.0}$ | $17.2_{\pm 6.0}$ |
| Open | Multiplan | $81.0_{\pm 7.8}$ | $20.6_{\pm 6.1}$ | $12.9_{\pm 4.9}$ | $22.2_{\pm 6.7}$ | $25.5_{\pm 6.3}$ | $17.8_{\pm 5.7}$ | $26.7_{\pm 6.8}$ |
| **GPT-5.1 Codex Mini** | | | | | | | | |
| Instructed | | $47.0_{\pm 10.0}$ | $34.4_{\pm 8.8}$ | $34.2_{\pm 9.3}$ | $34.7_{\pm 9.0}$ | $70.4_{\pm 7.0}$ | $75.6_{\pm 6.7}$ | $69.0_{\pm 7.6}$ |
| Open | Direct | $79.0_{\pm 8.1}$ | $7.2_{\pm 4.1}$ | $5.3_{\pm 4.0}$ | $7.5_{\pm 4.3}$ | $9.7_{\pm 4.6}$ | $7.5_{\pm 4.6}$ | $9.7_{\pm 4.7}$ |
| Open | Plan | $66.0_{\pm 9.4}$ | $9.8_{\pm 4.4}$ | $6.2_{\pm 4.1}$ | $10.5_{\pm 4.8}$ | $14.7_{\pm 5.0}$ | $9.8_{\pm 4.8}$ | $15.1_{\pm 5.4}$ |
| Open | Multiplan | $58.0_{\pm 9.8}$ | $4.2_{\pm 2.8}$ | $2.8_{\pm 3.0}$ | $4.7_{\pm 3.1}$ | $13.5_{\pm 5.0}$ | $9.7_{\pm 5.0}$ | $14.2_{\pm 5.4}$ |
| **Claude Sonnet 4.5** | | | | | | | | |
| Instructed | | $43.0_{\pm 9.9}$ | $32.1_{\pm 8.2}$ | $34.6_{\pm 9.2}$ | $31.4_{\pm 8.4}$ | $67.0_{\pm 6.1}$ | $79.0_{\pm 5.4}$ | $62.6_{\pm 7.3}$ |
| Open | Direct | $66.0_{\pm 9.4}$ | $6.2_{\pm 3.6}$ | $5.1_{\pm 3.6}$ | $6.4_{\pm 3.8}$ | $12.0_{\pm 4.7}$ | $10.8_{\pm 5.0}$ | $12.1_{\pm 5.0}$ |
| Open | Plan | $57.0_{\pm 9.9}$ | $10.7_{\pm 4.8}$ | $6.6_{\pm 4.3}$ | $11.7_{\pm 5.4}$ | $21.1_{\pm 6.2}$ | $16.8_{\pm 6.2}$ | $21.7_{\pm 6.7}$ |
| Open | Multiplan | $50.0_{\pm 10.0}$ | $10.1_{\pm 4.9}$ | $8.8_{\pm 5.2}$ | $10.7_{\pm 5.3}$ | $17.3_{\pm 5.5}$ | $17.4_{\pm 6.4}$ | $18.0_{\pm 6.1}$ |
| **MiniMax M2.7** | | | | | | | | |
| Instructed | | $49.0_{\pm 10.0}$ | $30.5_{\pm 7.8}$ | $36.4_{\pm 8.9}$ | $29.0_{\pm 8.0}$ | $62.8_{\pm 6.2}$ | $75.2_{\pm 5.5}$ | $58.9_{\pm 7.1}$ |
| Open | Direct | $71.0_{\pm 9.0}$ | $7.0_{\pm 4.0}$ | $3.3_{\pm 3.4}$ | $7.6_{\pm 4.3}$ | $11.1_{\pm 4.7}$ | $7.3_{\pm 4.4}$ | $11.5_{\pm 5.0}$ |
| Open | Plan | $63.0_{\pm 9.6}$ | $3.4_{\pm 2.6}$ | $4.6_{\pm 3.5}$ | $3.1_{\pm 2.6}$ | $10.7_{\pm 4.6}$ | $10.5_{\pm 4.8}$ | $10.8_{\pm 5.0}$ |
| Open | Multiplan | $58.0_{\pm 9.8}$ | $7.4_{\pm 4.0}$ | $4.5_{\pm 3.3}$ | $7.7_{\pm 4.4}$ | $15.3_{\pm 5.2}$ | $10.4_{\pm 4.7}$ | $15.9_{\pm 5.8}$ |
| **Qwen3** | | | | | | | | |
| Instructed | | $30.0_{\pm 9.1}$ | $10.7_{\pm 4.7}$ | $17.6_{\pm 6.8}$ | $8.7_{\pm 4.6}$ | $36.4_{\pm 5.6}$ | $56.6_{\pm 6.5}$ | $30.6_{\pm 6.1}$ |
| Open | Direct | $58.0_{\pm 9.8}$ | $2.0_{\pm 2.3}$ | $1.1_{\pm 1.1}$ | $2.2_{\pm 2.4}$ | $6.5_{\pm 3.5}$ | $4.4_{\pm 3.2}$ | $6.7_{\pm 3.7}$ |
| Open | Plan | $35.0_{\pm 9.5}$ | $2.1_{\pm 2.2}$ | $1.5_{\pm 1.3}$ | $2.2_{\pm 2.3}$ | $8.3_{\pm 3.9}$ | $4.1_{\pm 2.8}$ | $8.8_{\pm 4.4}$ |
| Open | Multiplan | $43.0_{\pm 9.9}$ | $2.2_{\pm 2.3}$ | $0.8_{\pm 1.0}$ | $2.6_{\pm 2.6}$ | $6.9_{\pm 3.3}$ | $5.3_{\pm 3.3}$ | $7.0_{\pm 3.6}$ |
| Golden Patch | | 100.0 | 100.0 | 100.0 | 100.0 | 100.0 | 100.0 | 100.0 |
| Base Commit | | 100.0 | 0.0 | 0.0 | 0.0 | 0.0 | 0.0 | 0.0 |

*Table 6.* Full experimental results across all tracks and models for PREC in percents with 95 % confidence intervals. We also report average filtered LoC edited $|L_{\hat{X}}|$ in patches.

| | | PREC (%) | PREC$^+$ (%) | PREC$^-$ (%) | $|L_{\hat{X}}|$ | $|L_{\hat{X}}^+|$ | $|L_{\hat{X}}^-|$ |
|---|---|---|---|---|---|---|---|
| GPT-5.2 | | | | | | | |
| Instructed | | $48.4 \pm 5.0$ | $41.2 \pm 6.3$ | $56.0 \pm 6.2$ | 1978.4 | 974.0 | 1004.5 |
| Open | Direct | $16.7 \pm 5.0$ | $7.6 \pm 4.1$ | $27.0 \pm 7.6$ | 632.0 | 303.8 | 328.2 |
| Open | Plan | $15.0 \pm 4.4$ | $6.1 \pm 3.6$ | $27.1 \pm 7.0$ | 1671.5 | 718.9 | 952.7 |
| Open | Multiplan | $19.8 \pm 4.8$ | $8.8 \pm 4.0$ | $36.8 \pm 7.6$ | 1539.1 | 811.4 | 727.8 |
| GPT-5.1 Codex Mini | | | | | | | |
| Instructed | | $50.2 \pm 5.6$ | $42.0 \pm 6.8$ | $58.5 \pm 6.5$ | 1045.8 | 510.0 | 535.8 |
| Open | Direct | $15.8 \pm 5.2$ | $7.0 \pm 4.6$ | $28.9 \pm 8.1$ | 331.2 | 140.9 | 190.2 |
| Open | Plan | $15.2 \pm 4.6$ | $4.9 \pm 3.2$ | $27.7 \pm 7.4$ | 811.7 | 419.8 | 391.9 |
| Open | Multiplan | $13.9 \pm 4.4$ | $4.7 \pm 3.5$ | $27.4 \pm 7.2$ | 544.1 | 296.6 | 247.5 |
| Claude Sonnet 4.5 | | | | | | | |
| Instructed | | $50.1 \pm 5.4$ | $39.2 \pm 6.3$ | $61.1 \pm 6.2$ | 1522.4 | 1011.2 | 511.2 |
| Open | Direct | $13.7 \pm 4.4$ | $6.8 \pm 4.1$ | $25.6 \pm 7.4$ | 929.4 | 462.1 | 467.3 |
| Open | Plan | $15.8 \pm 4.6$ | $9.3 \pm 4.4$ | $27.9 \pm 7.0$ | 2237.4 | 911.0 | 1326.3 |
| Open | Multiplan | $13.9 \pm 4.3$ | $7.6 \pm 3.9$ | $26.7 \pm 6.8$ | 2549.7 | 1905.0 | 644.6 |
| MiniMax M2.7 | | | | | | | |
| Instructed | | $52.8 \pm 5.6$ | $43.3 \pm 6.7$ | $61.4 \pm 6.4$ | 1000.2 | 539.7 | 460.5 |
| Open | Direct | $16.3 \pm 5.2$ | $7.8 \pm 4.7$ | $24.6 \pm 7.3$ | 293.8 | 127.2 | 166.6 |
| Open | Plan | $17.0 \pm 5.7$ | $9.7 \pm 5.3$ | $25.0 \pm 7.3$ | 297.5 | 152.4 | 145.1 |
| Open | Multiplan | $17.8 \pm 5.2$ | $9.7 \pm 4.9$ | $26.5 \pm 6.9$ | 571.9 | 369.3 | 202.6 |
| Qwen3 | | | | | | | |
| Instructed | | $47.3 \pm 6.4$ | $39.9 \pm 7.4$ | $51.7 \pm 7.2$ | 1130.5 | 784.5 | 346.1 |
| Open | Direct | $7.7 \pm 3.7$ | $5.3 \pm 3.8$ | $10.5 \pm 5.4$ | 1334.7 | 101.8 | 1232.9 |
| Open | Plan | $12.2 \pm 4.7$ | $6.1 \pm 4.2$ | $18.8 \pm 6.3$ | 730.9 | 298.1 | 432.8 |
| Open | Multiplan | $10.3 \pm 3.9$ | $4.0 \pm 3.4$ | $23.3 \pm 7.2$ | 440.1 | 183.1 | 257.0 |
| Golden Patch | | 48.5 | 42.4 | 54.1 | 1954.4 | 931.1 | 1023.3 |
| Base Commit | | – | – | – | 0.0 | 0.0 | 0.0 |

## A.6. Oracle Judge Validation

Oracle Multiplan relies on an LLM-based judge to select one candidate plan from a set of generated plans. To test whether this selection step introduces systematic bias, we run a randomized six-plan control for GPT-5.2 and SONNET 4.5. For each task, the judge is shown the five original Oracle Multiplan candidates together with the single plan from Plan mode, in a deterministic randomized order. We then map the selected shuffled index back to the original candidate identity.

The stored benchmark outputs contain observed IFR only for two candidates: the original single-plan execution ($P$) and the originally selected Multiplan execution ($BP$). They do not contain executions for the four Multiplan candidates that were generated but not selected. Therefore, we can directly measure judge accuracy only when the randomized judge selects either $P$ or $BP$. On non-tie cases where this comparison is evaluable, the judge selects the higher-IFR observed plan in 71.0% of cases overall (73.7% for GPT-5.2 and 68.0% for SONNET 4.5). If cases where the judge selects one of the four unobserved Multiplan candidates are counted as incorrect, the corresponding strict accuracy is 65.5% overall (68.9% for GPT-5.2 and 61.8% for SONNET 4.5).

This control also helps interpret the SONNET 4.5 Multiplan drop. For GPT-5.2, the full-benchmark average IFR is 16.4% in Plan mode, 25.5% in Oracle Multiplan, and 26.1% under the randomized six-plan judge-implied selection. For SONNET 4.5, the corresponding values are 21.1%, 17.3%, and 21.7%. Thus, when the single-plan candidate is available to the judge, the implied IFR recovers for SONNET 4.5 and remains at least as high for GPT-5.2. We therefore find no evidence that the Oracle Judge introduces systematic bias into plan selection; the lower SONNET 4.5 Multiplan score is more consistent with lower quality among the multiple generated plans than with a biased judge.

## A.7. Language Breakdown

Table 7 reports mean alignment score by primary programming language. We do not observe a consistent language-specific trend across settings.

*Table 7.* Mean alignment score by primary programming language. We report mean $\pm$ standard error over model-instance runs; lower scores indicate more difficult alignment. C includes C/C++ instances and has 3 benchmark instances.

| Setting | C | Go | Java | JavaScript | Python | Rust |
|---|---|---|---|---|---|---|
| Instructed | $57.5 \pm 11.8$ | $34.4 \pm 3.4$ | $34.7 \pm 5.0$ | $35.6 \pm 3.6$ | $30.2 \pm 4.9$ | $44.4 \pm 7.5$ |
| Open / Direct | $29.7 \pm 11.2$ | $6.4 \pm 1.4$ | $3.7 \pm 1.3$ | $5.9 \pm 1.5$ | $5.2 \pm 2.2$ | $1.5 \pm 1.1$ |
| Open / Plan | $29.0 \pm 10.6$ | $7.8 \pm 1.5$ | $1.8 \pm 1.1$ | $9.6 \pm 1.9$ | $8.4 \pm 2.8$ | $10.4 \pm 4.3$ |
| Open / Multiplan | $18.9 \pm 9.7$ | $8.7 \pm 1.7$ | $5.8 \pm 1.9$ | $9.8 \pm 2.0$ | $8.7 \pm 2.8$ | $9.8 \pm 3.6$ |

### A.8. Pairwise Significance Tests

We test pairwise differences between agents within each track and inference mode using two-sided paired sign-flip permutation tests over shared instances, with 100,000 Monte Carlo sign-flip permutations per comparison. We apply Benjamini–Hochberg correction across all pairwise comparisons for the same metric, using $q \leq 0.05$ as the significance criterion. Table 8 summarizes corrected-significant wins and losses for each metric, track, inference mode, and model. All comparisons use the same 100 paired instances.

*Table 8.* Pairwise significance summary. Each model cell reports corrected-significant wins/losses against the other models in the same metric, track, and mode.

| Metric | Track | Mode | GPT-5.2 | GPT-5.1 M | SONNET 4.5 | MINIMAX M2.7 | QWEN3 |
|---|---|---|---|---|---|---|---|
| IFR | Instructed | | 4/0 | 2/1 | 1/1 | 1/2 | 0/4 |
| | Open | Direct | 0/0 | 0/0 | 1/0 | 0/0 | 0/1 |
| | Open | Plan | 1/0 | 1/1 | 3/0 | 0/1 | 0/3 |
| | Open | Multiplan | 4/0 | 1/1 | 1/1 | 1/1 | 0/4 |
| PASS | Instructed | | 4/0 | 1/1 | 1/1 | 1/1 | 0/4 |
| | Open | Direct | 3/0 | 2/0 | 0/2 | 0/1 | 0/2 |
| | Open | Plan | 4/0 | 1/1 | 1/1 | 1/1 | 0/4 |
| | Open | Multiplan | 4/0 | 1/1 | 0/1 | 1/1 | 0/3 |
| $\mathcal{A}$ | Instructed | | 4/0 | 1/1 | 1/1 | 1/1 | 0/4 |
| | Open | Direct | 1/0 | 0/0 | 0/0 | 0/0 | 0/1 |
| | Open | Plan | 2/0 | 2/0 | 2/0 | 0/3 | 0/3 |
| | Open | Multiplan | 4/0 | 0/2 | 2/1 | 1/1 | 0/3 |

## B. Static Rule Examples

This section shows an additional rule example focused on symbolic propagation to isolate the matching mechanics of a single rule.

```
rule γ
1  pattern: $THEME.colors.$PROP
2  options:
3    symbolic_propagation: true
```

```
file f
1  const themeColors = theme.colors;
2  const primaryColor = themeColors.primary; // ω₁
3  const otherColors =
4    (theme.colors.secondary, theme.colors.ternary); // ω₂, ω₃
```

*Figure 9.* Example OpenGrep rule $\gamma$ with symbolic propagation on repository $R = \{f\}$. Colored tokens show metavariable bindings, e.g., $THEME denotes the bound theme object, while green line highlights show line coverage $\mathcal{C}$. The witnesses are $\Omega(\gamma, R) = \{\omega_1, \omega_2, \omega_3\}$, so the match count is $\mathcal{M}(\gamma, R) = 3$; the coverage contains two source lines because $\omega_2$ and $\omega_3$ occur on the same line.

# C. Open Track Case Studies

In this section, we present case studies comparing agent outputs against the ground truth refactorings. For each case, we use `claude-code` with OPUS 4.6 to assist us in parsing the generated patches obtained from GPT-5.2 and SONNET 4.5, their respective evaluation artifacts, and comparing them to the ground truth $X^*$. We then manually review the resulting report, verify the claims, and present the following takeaways. The prompt used in this part can be found in Figure 20.

## C.1. Mockito

The refactoring restructures Mockito's flat `org.mockito.internal` package into five sub-packages (`creation`, `invocation`, `state`, `stubbing`, `verification`), renames several core classes and methods (e.g., `VerifyingMode` to `OngoingVerifyingMode`, `andThrows` to `andThrow`), and updates all imports and cross-references accordingly. The changes span approximately 2,000 lines across the core Mockito source. The precise commit is `mockito/mockito@2f7bf91d`.

### C.1.1. OPEN TRACK

The task provided to the agents is: `Improve internal organization and naming`.

**Direct Mode.** In direct mode, none of the agents proceeds to perform a refactoring that resembles the scope of $X^*$, instead they are attracted to fixing tiny, surface-level nitpicks. They preserve and validate functional correctness.

SONNET 4.5 renamed typo-containing hamcrest matcher classes and removed `I` prefixes from interface names (`IArgumentMatcher` to `ArgumentMatcher`) and runs tests. It achieved an IFR of $0.98\%$, by coincidentally removing the `IAnswer` file.

GPT-5.2 fixed the same typos using a backwards-compatible deprecated-wrapper approach, replaced a wildcard import, cleaned up some parameter names, and runs tests. It performed none of the structural package moves or renames contained in the ground-truth. It achieved an IFR of $0.0\%$.

**Plan Mode.** Both agents resort to a similar set of superficial edits. Both successfully run tests and builds.

SONNET 4.5 focuses on renaming typo-containing classes. Destructive file renames created duplicate test classes. No structural changes were made. It achieved an IFR of $0.0\%$.

GPT-5.2 focuses on fixing typos in class names and improving overall naming. No structural changes were made. It achieved an IFR of $0.98\%$ by coincidentally renaming the field `mocksToBeVerifiedInSequence` to `mocksToVerifyInOrder`.

**Multiplan Mode.** The edits are no longer purely superficial, but address fundamental code quality concerns. Each of the plans created by SONNET 4.5 addresses a basket of issues, resulting in incomplete edits. The plans proposed by GPT-5.2 are more focused resulting in a consistent and correct patch.

SONNET 4.5 achieved an IFR of $4.9\%$, the agent addressed state package extraction (1/5 subpackages), performed no type/method renames, yet introduced new unnecessary abstractions (e.g., StateManager interface). The introduction of a partial field update results in missing field errors, causing tests and builds to fail. The agent is not able to recover from this inconsistent state.

GPT-5.2 achieved an IFR of $22.5\%$. The agent successfully reorganized all 5 target subpackages with correct import/package updates across 60+ files, but did not attempt any of the type or method renames that account for the remaining $50\%$ of positive rules. The agent successfully builds the updated codebase.

### C.1.2. INSTRUCTED TRACK

Both agents address the specifications provided in the detailed task description, resulting in a perfect or near perfect IFR of $99\%$ and $100\%$ for SONNET 4.5 and GPT-5.2, respectively. GPT-5.2 uses targeted patches for its edits, resulting in a successful build. SONNET 4.5 makes extensive use of risky replace all `sed -i` commands on all `.java` files in the `test` directory. This results in an inconsistent state and build errors.

## C.2. AWS CLI

The refactoring migrates the `aws-cli` test suite from the third-party `mock==1.3.0` package to Python's built-in `unittest.mock`, updating `awscli/testutils.py` and `tests/__init__.py` as centralized re-export points and rewriting imports across 178 test files to use project-idiomatic `from tests import mock` or `from awscli.testutils import mock` patterns.

The precise commit is `aws/aws-cli@20315462`.

### C.2.1. OPEN TRACK

The task provided to the agents is: `Update test mocking dependencies`.

**Direct Mode.** SONNET 4.5 achieved an IFR of $15.6\%$ by correctly updating the two infrastructure files and 144 of 178 test files, but used `from unittest import mock` directly in each file instead of the ground-truth's project-idiomatic `from tests import mock` / `from awscli.testutils import mock` convention. The generated patch $\hat{X}$ is not complete, however the IFR also punishes that the agent did not use the `from tests import mock` convention, which is a bit harsh.

GPT-5.2 achieved an IFR of $3.1\%$ by updating only the two infrastructure files and creating a `sys.modules` compatibility shim (`tests/mock.py`) to make old `import mock` statements work transparently, rather than actually migrating any test file imports; the near-zero score accurately reflects that the agent bypassed the refactoring entirely, leaving all 178 test files unchanged with their old-style import patterns intact.

**Plan Mode.** Both agents perform a similar set of edits as in the reference refactoring $X^*$. SONNET 4.5 validates its edits by running tests, GPT-5.2 only validates the syntax using `python -m compileall`. The achieved IFR scores are $60.9\%$ and $81.3\%$ for GPT-5.2 and SONNET 4.5, respectively. The two resulting codebases pass our test suite.

The behavior in **Multiplan Mode** and **Instructed Track** is almost identical. Both agents achieve $81.3\%$, the non-perfect score stems from using `from awscli.testutils import mock` instead of `from tests import mock`.

## C.3. gqlgen

The refactoring modernizes the entire gqlgen Go codebase by replacing all occurrences of `interface{}` with Go 1.18's built-in `any` alias, a purely mechanical refactoring affecting files across library code, templates, tests, and examples, with no behavioral changes. The precise commit is `99designs/gqlgen@d5c9f896`.

### C.3.1. OPEN TRACK

The task provided to the agents is `Modernize type declarations across the codebase`.

**Direct Mode.** Both agents perform the refactoring to a substantial degree.

SONNET 4.5 achieved an IFR of $93.3\%$, successfully replacing `interface{}` with any across approximately 193 files and passing all 17 additive rules, but left behind `interface{}` in a small number of locations (6 out of 73 reductive rule violations), including some variable declarations, unmarshal function parameters, and function return types.

GPT-5.2 achieved a perfect IFR of $100.0\%$, comprehensively replacing every `interface{}` occurrence with any across 231 files with zero remaining legacy patterns detected, fully accomplishing the refactoring.

**Plan Mode.** SONNET 4.5 achieved IFR of $63.3\%$ by replacing `interface{}` with any in core runtime, client, config, handler, and plugin source files. It ran `go test` repeatedly, but did not run `go generate` to propagate changes to generated code, leaving residues in generated files. GPT-5.2 achieved IFR of $78.9\%$ by replacing `interface{}` with any across source files and running `go generate ./...` to propagate template changes to generated code. It never ran tests and exhausted its $11 budget before completion. The resulting patch $\hat{X}$ contains genuine gaps in hand-written files (e.g., `complexity/` package, various Unmarshal signatures) that the agent did not reach before budget exhaustion.

**Multiplan Mode.** SONNET 4.5 achieved an IFR of $100\%$ by performing a comprehensive repository-wide `sed`-based replacement of `interface{}` to any across 232 files, and running tests successfully multiple times.

GPT-5.2 achieved an IFR of $100\%$ via a `perl`-based global replacement followed by `go generate` across 231 files, and preserved functional correctness. It claims to have run tests, but actually did not.

## C.3.2. INSTRUCTED TRACK

SONNET 4.5 achieved IFR of $100\%$ by performing a comprehensive `sed -i`-based global replacement of `interface{}` with `any` across all `.go` files. It adds a linter rule and verifies correctness via `go build` and `go test`.

GPT-5.2 achieved IFR of $96.7\%$ using `perl` and `gofmt -r` to correctly handle most replacements. Three residual occurrences of `interface{}` in parameter/field declarations remain, however. These violations were not captured by `gofmt -r`. The test suite is passing even though the agent never actually runs `go test` or `go build`.

## C.4. Bytebase

The refactoring standardizes Vue.js route constant naming across 51 files by adding a `_ROUTE_` infix to project route constants, extracting workspace setting route names from hardcoded strings into exported constants, refactoring sidebar components for route-name-based navigation, and creating a reusable `WebhookTypeIcon` component. The precise commit is `bytebase/bytebase@04ac644e`.

## C.4.1. OPEN TRACK

**Direct Mode.**    The task provided to the agents is `Refactor routing system`.

SONNET 4.5 refactored the Go backend server routing (middleware extraction, service registry pattern) rather than the intended Vue.js frontend route constants, achieving an IFR of $0.0\%$.

GPT-5.2 removed the legacy `routeSlug` method and `RouterSlug` type from the frontend router store, which is completely orthogonal to the constant naming standardization addressed in the human refactoring. It achieved an IFR of $0.0\%$.

The plans proposed in **Plan Mode** and **Multiplan Mode** are larger in scope, but remain orthogonal to the frontend refactoring $X^*$. The IFR achieved for both agents in both modes remains at $0.0\%$.

## C.4.2. INSTRUCTED TRACK

SONNET 4.5 achieved IFR $100\%$ by successfully applying the core route-constant renaming and setting-route centralization across 43 files. This is 8 fewer files than the golden patch $X^*$. The IFR of $100\%$ is inflated due to binary rule satisfaction constraints as defined in §4.1. The agent uses broad `sed -i` commands, resulting in much lower cost compared to GPT-5.2.

GPT-5.2 achieved IFR $100\%$ by applying the refactoring more broadly than $X^*$, touching 87 files (vs 51 in $X^*$), adding route constants in files beyond the scope of $X^*$. The agent proceeded using many targeted patches until finally hitting the budget constraint.

# D. Prompts

In this section, we detail all prompts used for the respective models and tasks. The prompts include: (i) scoring prompts used in the data creation and filtering pipeline (Figures 10 and 11), (ii) task generation prompts for the instructed and open tracks (Figures 14 and 15), and (iii) prompts used during evaluation for planning, execution, and oracle plan selection (Figures 12, 13 and 16–19).

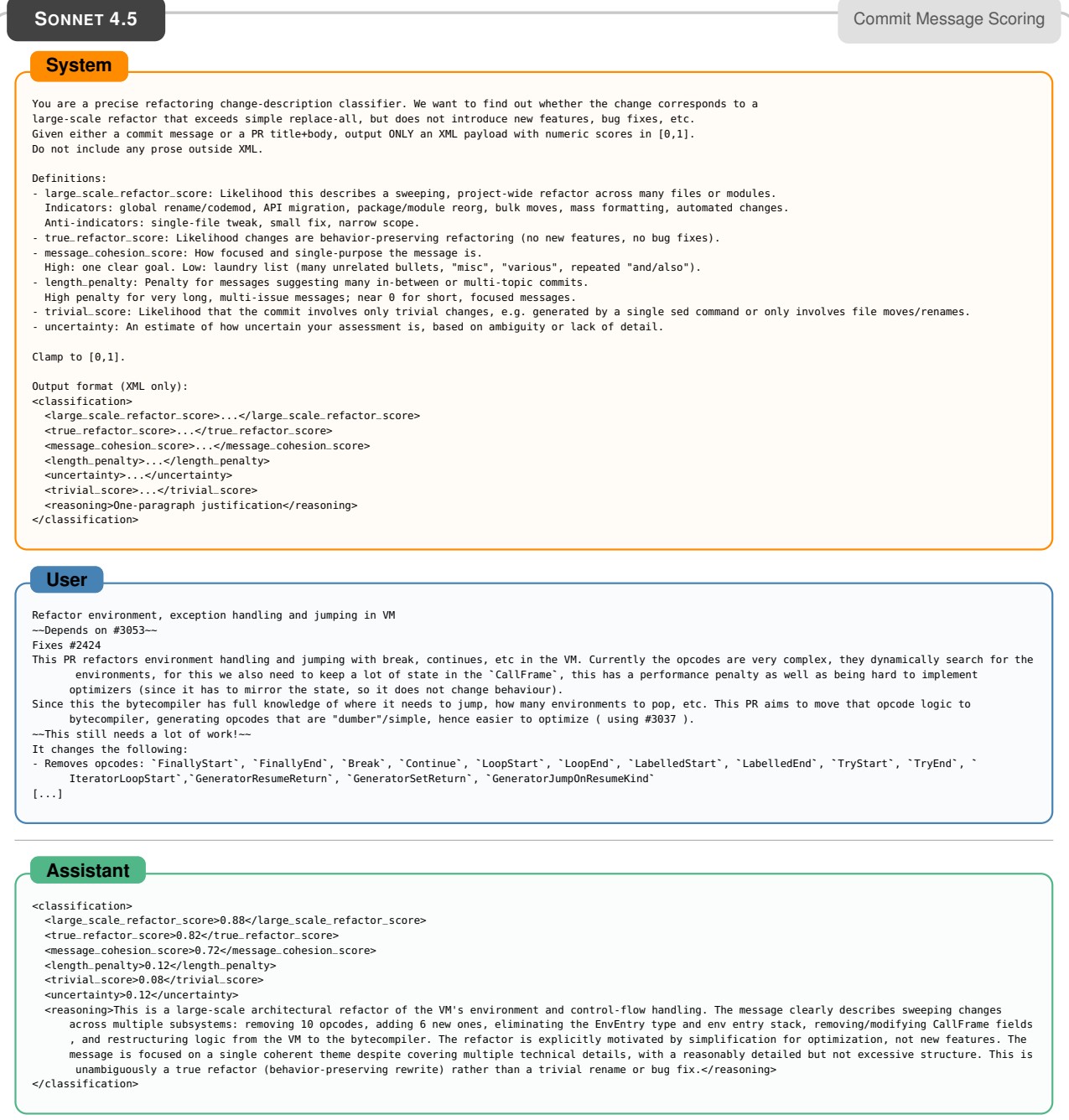

*Figure 10.* Commit message scoring prompt used as a late-stage filter in the data scraping funnel.

**SONNET 4.5**                                                                                    Diff Scoring

**System**

```
Your task is to identify whether the unified diff provided is a true, useful large-scale refactoring commit.
Given a unified diff (git format), output ONLY an XML payload with numeric scores in [0,1].
Do not include any prose outside XML.
Definitions and guidance:
- refactor_likelihood: Likelihood changes are behavior-preserving refactoring (rename/move, code reorganization, function extraction/inlining, variable/method
      renames, import/namespace changes, signature reshaping, mass formatting).
- mixed_change_penalty: Penalty if the diff mixes refactors with features/bugfixes or risky changes.
- risky_change_score: Evidence of behavior-altering edits (new logic, feature flags, new APIs).
- automated_pattern_score: Signs that refactoring results from automated tools, linters or similar.
- uncertainty: Reflects ambiguity, lack of context, or insufficient signal in the diff.
- complexity: Give an estimate of how complex this refactor is. (0=trivial, 0.5=moderate, 1=extremely complex)
- trivial_score: Indicates the presence of trivial changes (e.g., whitespace, comments).

Output format (XML only):
<classification>
  <refactor_likelihood>...</refactor_likelihood>
  <mixed_change_penalty>...</mixed_change_penalty>
  <risky_change_score>...</risky_change_score>
  <automated_pattern_score>...</automated_pattern_score>
  <uncertainty>...</uncertainty>
  <reasoning>One-paragraph justification</reasoning>
  <complexity>...</complexity>
  <trivial_score>...</trivial_score>
</classification>
```

**User**

```
diff --git a/core/engine/src/builtins/eval/mod.rs b/core/engine/src/builtins/eval/mod.rs
index 3e6c905e5fd..55b38f31405 100644
--- a/core/engine/src/builtins/eval/mod.rs
+++ b/core/engine/src/builtins/eval/mod.rs
@@ -20,7 +20,7 @@ use crate::{
     string::StaticJsStrings,
-    vm::{CallFrame, CallFrameFlags, Constant, Registers},
+    vm::{CallFrame, CallFrameFlags, Constant},
@@ -333,8 +333,7 @@ impl Eval {
-        let register_count = context.vm.frame().code_block().register_count;
-        let record = context.run(&mut Registers::new(register_count as usize));
+        let record = context.run();
         context.vm.pop_frame();
[...]
```

**Assistant**

```
<classification>
  <refactor_likelihood>0.92</refactor_likelihood>
  <mixed_change_penalty>0.05</mixed_change_penalty>
  <risky_change_score>0.25</risky_change_score>
  <automated_pattern_score>0.75</automated_pattern_score>
  <uncertainty>0.08</uncertainty>
  <reasoning>This is a large-scale, systematic refactoring that consolidates VM register and stack management. The `Registers` struct is eliminated and its
      functionality is merged into a new `Stack` struct. All ~50+ opcode handler functions are consistently updated to remove the `&mut Registers` parameter and
      access registers through `context.vm` instead. CallFrame methods are refactored to return indices rather than values. The changes are behavior-preserving
      reorganization with very consistent patterns across all files, showing evidence of automated tooling supplemented by manual coordination. While the
      structural changes are deep and touch core VM systems (register access, promise capabilities, async generators), the refactoring appears carefully
      coordinated with no obvious feature additions or bug fixes mixed in.</reasoning>
  <complexity>0.85</complexity>
  <trivial_score>0.12</trivial_score>
</classification>
```

*Figure 11.* Diff scoring prompt used to rank candidate commits by refactoring quality.

## CODETASTE: Can LLMs Generate Human-Level Code Refactorings?

SONNET 4.5

**System**

```
## Your Environment

You operate in a containerized, non-interactive **polyglot development environment** based on **Ubuntu 24.04**.

### **Core Runtimes**

* **Python:** Managed by `uv` (versions **3.8, 3.9, 3.10, 3.11**, see `uv python list`).
* **Node.js:** **v22.12.0** (via NVM `nvm`) with **TypeScript**, `ts-node`, and `vercel`.
* **Go:** **v1.23.4** (Global toolchain).
* **Rust:** Full toolchain (Cargo/Rustup) in `/opt/rust`.
* **.NET:** **SDK 8.0** (LTS).
* **C/C++:** GCC, Clang, CMake, and `build-essential`.
* **Java:** OpenJDK (`default-jdk`).

### **Pre-installed Tools**

* **Testing:** **Playwright** (with Chromium and system deps).
* **System:** `git`, `curl`, `wget`, `vim`, `sudo`, and build libraries (SSL, ffi, sqlite).

### **Environment Specs**

* **User:** `benchmarker` (non-root with passwordless `sudo`).
* **Repository Directory:** `/testbed`. `/testbed` is wiped on exit.
* **Key Paths:** Tools are pre-configured in `$PATH` (Node, Rust, .NET, Go, uv).
## Task

Configure the development environment and validation scripts for the repository located in `/testbed/`.

**Constraints:**

1. **EXPLORE:** Analyze the `/testbed/` directory to identify the primary programming language, required runtime versions (e.g., via version files or manifests), the preferred
    package manager, and the testing framework used.
2. **DEPS:** Identify and install necessary system-level dependencies using `sudo` (non-interactive) and all project-level dependencies. Ensure any external binaries or drivers
    required by the test suite (e.g., browser engines, compilers, or database headers) are installed immediately. Anything installed inside `/testbed/` will be wiped on exit!
3. **INTEGRITY:** NEVER modify files in /testbed/ directly, any changes you perform will be wiped after you exit! `/scripts/setup_shell.sh` can setup /testbed directory, however it
    should NOT modify versioned files in `/testbed/`, i.e. it should only modify files or folders that are explicitly ignored by the version control system (e.g., build artifacts,
    dependency directories, cache). `git status` must show no changes.
4. **SCRIPTS:**
* **Create `/scripts/setup_system.sh`:**
    Executed with `sudo` prior to running the tests, this script performs runtime system configuration (e.g., starting database services, Redis, or configuring system limits).
    It should **not** install packages. If no system services are required, create a script that exits 0.
* **Create `/scripts/setup_shell.sh`:**
    When sourced, this script configures the shell environment for the project **and** to run tests:
            - activate virtual environment if necessary.
            - install local project dependencies and linters.
            - set up all environment variables.
            - selects the correct runtime versions.
        - Project and dependency installations that change files in `/testbed/` MUST be performed in this script.
    It must **NOT** require `sudo`. It must be idempotent (safe to run multiple times) and avoid redundant installations.
* **Create `/scripts/run_tests`:** This script must be self-contained and execute the test suite (or a relatively large and representative subset that finishes in up to 15 minutes).
        It will be invoked as follows: `git clean -xdff && sudo /scripts/setup_system.sh && source /scripts/setup_shell.sh && /scripts/run_tests`.
    `/scripts/run_tests` must NOT setup the environment, environment variables, shell or similar, it should rely on the previous scripts having been run before. `/scripts/run_tests`
        must only run and parse tests. It should be invoked like : `/scripts/setup_shell.sh && /scripts/run_tests` to run tests in an already setup shell. `/scripts/setup_shell.sh`
        sets up the environment and environment variables.
* **Portability:** These scripts **must remain functional** even if `/testbed/` is checked out to THE previous commit (in other words: `HEAD−1`). I.e. they should work on HEAD and
        HEAD−1 without any modifications.
5. **OUTPUT:** The `/scripts/run_tests` script must output exactly one JSON line as its final `stdout` message:
`{"passed": int, "failed": int, "skipped": int, "total": int}`.
Where the int value corresponds to the number of tests that passed, failed, were skipped, and the total number of representative tests run.
You may need to write a wrapper or use a custom test reporter to ensure this exact format is captured. Test results must be **deterministic** and reflect actual test outcomes.
6. **VERIFICATION:** Always execute `source /scripts/setup_shell.sh && /scripts/run_tests` in a single shell command (in an environment that is clean -> `git clean -xdff` and has
        system services running `/scripts/setup_system.sh`) on both the current commit and `HEAD−1`. Confirm the JSON outputs accurately reflect the test results.
Your shell doesn't persist environment across shell commands.

**Note:** Multiple runtimes (Python, Node, Rust, .NET, Java, GO) and their respective version managers are pre-installed. Any global configurations or runtime versions you install
        will be preserved across the session.
**Note:** You run in a non-interactive terminal; Proceed with these actions without asking for further permission. Take action and perform the installation and changes.

## Final Notes
Once you have completed the task, briefly write a concise summary of the testing setup in `/scripts/SUMMARY.md` of the form:
# Summary
...
## System Dependencies
...
## PROJECT Environment
...
## Testing Framework
...
## Additional Notes
... (e.g. any obstacles or misconfigurations of the environment you encountered)
```

*Figure 12.* Setup Prompt for Environment Configuration

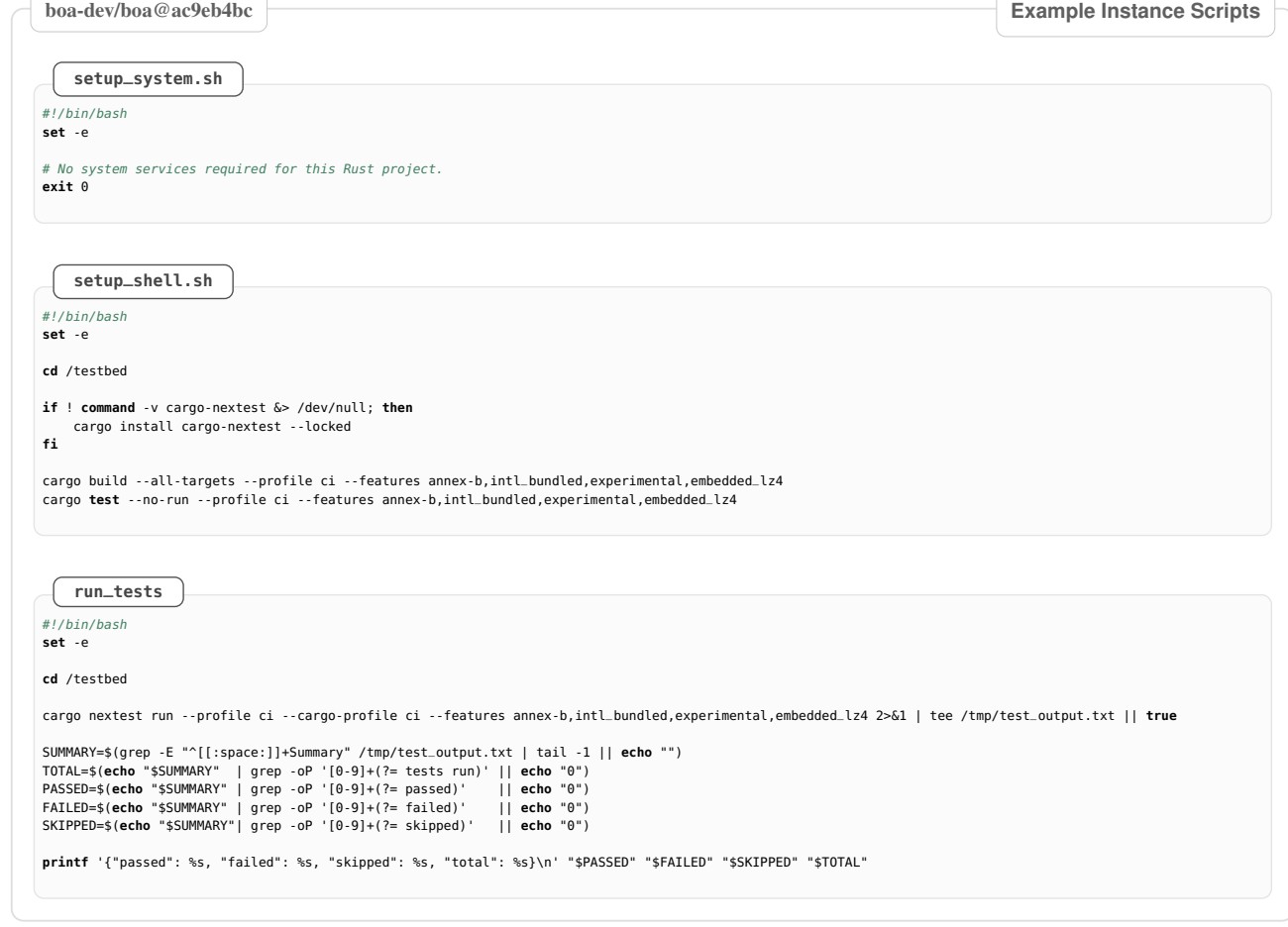

*Figure 13.* Excerpt of the standardized scripts generated during the setup phase for the boa-dev/boa instance (base commit ac9eb4bc).

**SONNET 4.5**                                                                                          Task Generation (Instructed Track)

### System

```
Generate a refactoring task description for the commit described by the user. It should be formatted like a GitHub issue.
Restrict yourself to the scope of this commit, do not talk about Follow-Up Tasks, Future or Past Work.
Only output the issue description, nothing else. Do not include any explanations.

[...]

Output Format should be well structured:
Title
-----
<title>
Summary
-------
<summary>
Why
---
<why>
[...]
```

### User

```
<repo_name>boa-dev/boa</repo_name>
<commit_message>Refactor registers to use the stack (#4263)
* Refactor registers to use the stack
* Fix import in doc comment</commit_message>
<pull_requests><pull_request>
<pr_title>Refactor registers to use the stack</pr_title>
<pr_body>This Pull Request changes the following:
- Refactor the vm stack into a dedicated struct and implement stack access in functions that are more readable.
- Remove the `Registers` data structure and store registers on the stack.
</pr_body></pull_request>
</pull_requests>
<commit_diff>diff --git a/core/engine/src/builtins/eval/mod.rs b/core/engine/src/builtins/eval/mod.rs
index 3e6c9..55b38 100644
--- a/core/engine/src/builtins/eval/mod.rs
+++ b/core/engine/src/builtins/eval/mod.rs
@@ -22,3 +22,3 @@ use crate::{
     string::StaticJsStrings,
-    vm::{CallFrame, CallFrameFlags, Constant, Registers},
+    vm::{CallFrame, CallFrameFlags, Constant},
     Context, JsArgs, JsResult, JsString, JsValue, SpannedSourceText,
@@ -335,4 +335,3 @@ impl Eval {
-        let register_count = context.vm.frame().code_block().register_count;
-        let record = context.run(&mut Registers::new(register_count as usize));
+        let record = context.run();
[...]</commit_diff>
```

### Assistant

```
# Title
-----
Refactor registers to use the stack

# Summary
-------
Refactor VM execution to store registers directly on the stack instead of maintaining a separate `Registers` data structure. Consolidate stack and register
    management into a single `Stack` type with improved API and readability.

# Why
---
- Eliminate duplication between stack and registers storage
- Improve memory efficiency by using single unified structure
- Better encapsulation with dedicated `Stack` type and semantic accessor methods
- Clearer separation between frame pointer, arguments, and register spaces

# Changes
---

**Removed `Registers` struct:**
- Deleted standalone `Registers` data structure (~50 lines)
- Removed `registers.push_function()` / `registers.pop_function()` calls
- No more `Registers::new()` or `registers.clone_current_frame()`

**New `Stack` struct:**
- Encapsulates `Vec<JsValue>` with ~200 lines of accessor methods
- Methods: `get_this()`, `get_function()`, `get_arguments()`, `get_register()`, `set_register()`, ...
- Calling convention helpers: `calling_convention_pop_arguments()`, `calling_convention_push_arguments()`, ...
- Frame management: `truncate_to_frame()`, `split_off_frame()`
- Special register accessors: `get_promise_capability()`, `async_generator_object()`, ...

[...]
```

*Figure 14.* Task description generation prompt used to synthesize an issue-style refactoring task from commit associated metadata.

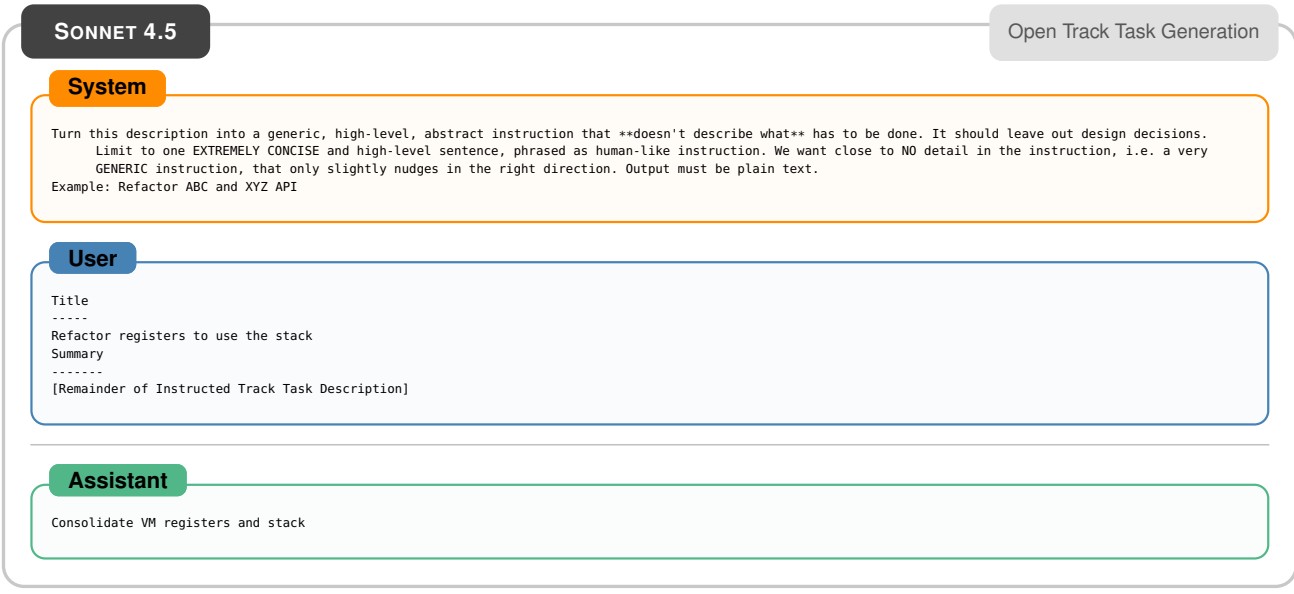

*Figure 15.* Prompt used to derive the open-track instruction by abstracting away actionable detail from an instructed task description.

SONNET 4.5                                                                 Open Track Task Generation

**System**

Turn this description into a generic, high-level, abstract instruction that **doesn't describe what** has to be done. It should leave out design decisions.
    Limit to one EXTREMELY CONCISE and high-level sentence, phrased as human-like instruction. We want close to NO detail in the instruction, i.e. a very
    GENERIC instruction, that only slightly nudges in the right direction. Output must be plain text.
Example: Refactor ABC and XYZ API

**User**

```
Title
-----
Refactor registers to use the stack
Summary
-------
[Remainder of Instructed Track Task Description]
```

**Assistant**

Consolidate VM registers and stack

ANY AGENT                                                                  Open Track: Plan Prompt

```
Conduct IN-DEPTH EXPLORATION and ANALYSIS of the codebase.
Refine the following task description into a CONCRETE and ACTIONABLE refactoring plan. Make ALL the design choices. DO NOT propose broad, multi-stage changes.
You operate completely AUTONOMOUSLY in this sandboxed environment. DO NOT ASK FOR CLARIFICATIONS.
The final plan must be in markdown format:
```
# Title
<Summary>
## Changes
...
## Why
...
```
You MUST PLACE the final plan in this file: '/output/refactoring_plan.md'.

Task:
[Task Description]
```

*Figure 16.* Prompt used to generate a single concrete refactoring plan from an open-track instruction.

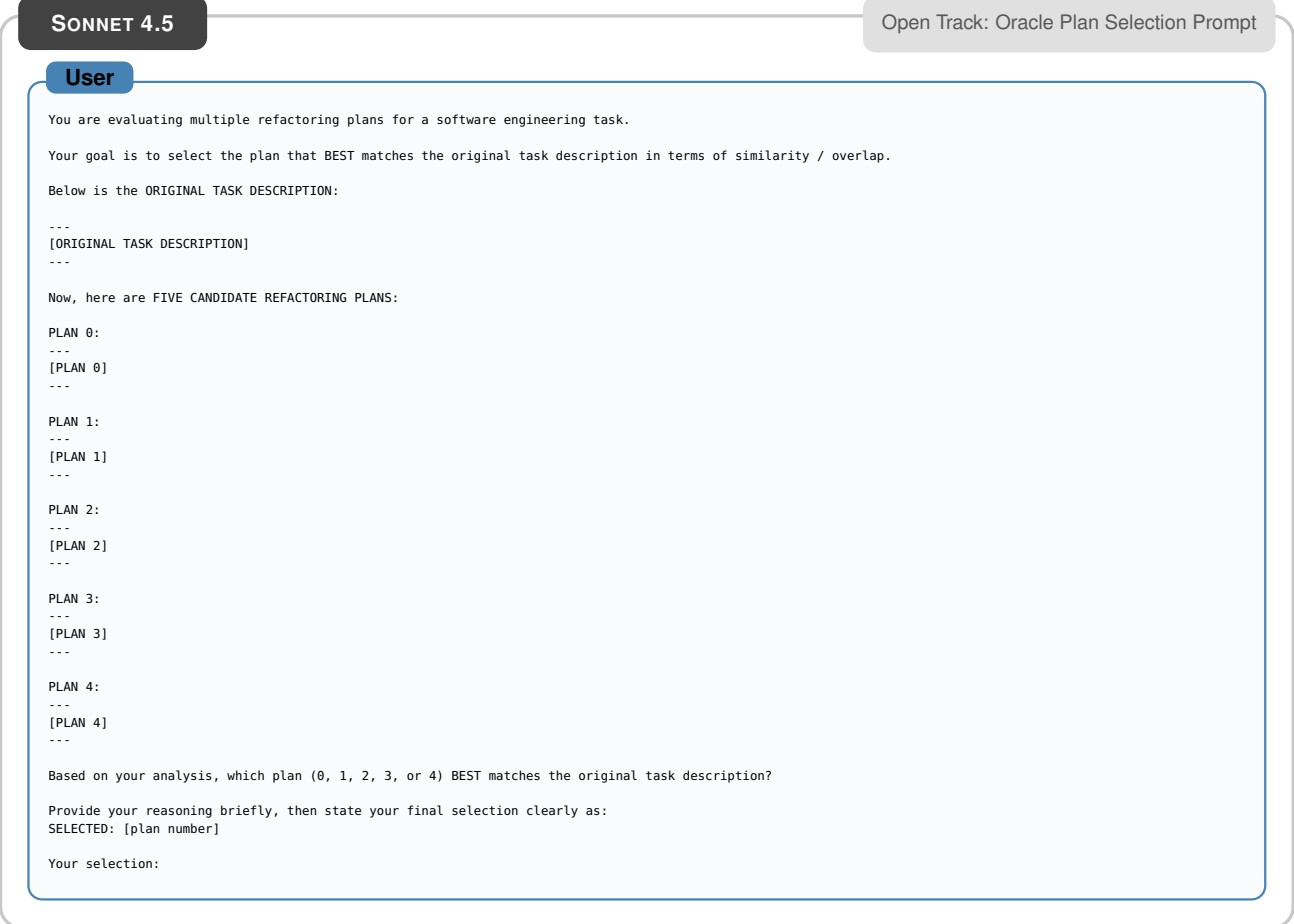

**ANY AGENT**

Open Track: Oracle Multiplan Prompt

```
Conduct IN-DEPTH EXPLORATION and ANALYSIS of the codebase.
Generate 5 DISTINCT and DIFFERENT refactoring plans for the following task. There can be partial overlap.
You operate completely AUTONOMOUSLY in this sandboxed environment. DO NOT ASK FOR CLARIFICATIONS.

For each approach, create a COMPLETE, CONCRETE and ACTIONABLE refactoring plan. Make ALL the design choices. DO NOT propose broad, multi-stage changes.
Each plan must be in markdown format:
```
# Title
<Summary>
## Changes
...
## Why
...
```

- '/output/refactoring_plans/refactoring_plan0.md'
- '/output/refactoring_plans/refactoring_plan1.md'
- '/output/refactoring_plans/refactoring_plan2.md'
- '/output/refactoring_plans/refactoring_plan3.md'
- '/output/refactoring_plans/refactoring_plan4.md'

Task:
[Task Description]
```

*Figure 17.* Prompt used to generate multiple distinct candidate refactoring plans from an open-track instruction.

**SONNET 4.5**

Open Track: Oracle Plan Selection Prompt

**User**

```
You are evaluating multiple refactoring plans for a software engineering task.

Your goal is to select the plan that BEST matches the original task description in terms of similarity / overlap.

Below is the ORIGINAL TASK DESCRIPTION:

---
[ORIGINAL TASK DESCRIPTION]
---

Now, here are FIVE CANDIDATE REFACTORING PLANS:

PLAN 0:
---
[PLAN 0]
---

PLAN 1:
---
[PLAN 1]
---

PLAN 2:
---
[PLAN 2]
---

PLAN 3:
---
[PLAN 3]
---

PLAN 4:
---
[PLAN 4]
---

Based on your analysis, which plan (0, 1, 2, 3, or 4) BEST matches the original task description?

Provide your reasoning briefly, then state your final selection clearly as:
SELECTED: [plan number]

Your selection:
```

*Figure 18.* Prompt used by the oracle judge to select the candidate plan that best matches the original refactoring description.

**CODETASTE: Can LLMs Generate Human-Level Code Refactorings?**

---

**ANY AGENT**                                                           Inference Prompt

```
Perform the task described below in it's ENTIRETY. You operate completely AUTONOMOUSLY in a sandboxed environment. DO NOT ASK FOR
    CLARIFICATIONS. You must EDIT the codebase DIRECTLY to complete the task. DO NOT create reports, plans or similar files.

[Task Description]
```

*Figure 19.* Default Prompt used to run inference on an agent.

---

**OPUS 4.6**                                                    `claude-code` Judge Prompt

```
You should create a report comparing the outputs of the two agents (`claude-code-v2.0.76-sonnet45`, `codex-v0.77.0-gpt-5.2`) to
    the ground-truth refactoring (`golden`).
Ignore .venv, .node_modules, .cargo-home and binary file edits in your report.
Give an assesment of whether the score regarding IFR is justified based on the evidence. I.e. does it give an accurate reflection
    on how much of the true reference refactoring the agent achieved.
Additionally, look at the agents performance regarding functional correctness by investigating the test results.
Connect the analysis to the inference.out to answer obvious questions, such as did the agent try to run tests or build?
Talking or thinking about running a command does not count as evidence. Only the actual commands and their outputs count.

Explore all files in this folder / subfolders to find evidence for your report, but only edit the `case_study_report.md` file.
Place the report inside the current directory. It should be a markdown file named `case_study_report.md`.
```md
[Evidence, Reasoning, Claims]

## Dense Summary

The refactoring [... 1-2 sentence summary about what the ground-truth refactoring achieves.]

claude-code-v2.0.76-sonnet45: [1-2 sentences about what the agent achieved, what IFR score is achieved (why and whether this is
    reflective), and did the agent preserve functional correctness? Why / why not? Also say what wasn't achieved if relevant.]
codex-v0.77.0-gpt-5.2: [1-2 sentences about what the agent achieved, what IFR score is achieved (why and whether this is
    reflective), and did the agent preserve functional correctness? Why / why not? Also say what wasn't achieved if relevant.]
```
```

*Figure 20.* Prompt used to run `claude-code` to assist evaluation of the different tracks in the case studies.

