# OpenReview forum: "CodeTaste: Can LLMs Generate Human-Level Code Refactorings?"
_ICML.cc/2026/Conference — ICML 2026 regular_

### Official Review · Reviewer_wheP · 2026-02-28

**Soundness:** 3
**Presentation:** 3
**Significance:** 3
**Originality:** 3
**Overall Recommendation:** 4
**Confidence:** 4

**Summary:**

The paper introduces CodeTaste, a new benchmark designed to evaluate the capability of LLM coding agents to perform large-scale, multi-file code refactorings. The authors develop an automated pipeline that mines 100 complex refactoring commits from GitHub, constructs reproducible containerized execution environments, and uses an LLM to generate static analysis rules to verify the semantic intent of the refactorings. The evaluation is split into an Instructed Track and an Open Track. Empirical findings reveal that while frontier models can reliably execute explicitly detailed refactorings, they struggle to autonomously discover human-aligned refactoring choices when provided with limited guidance.

**Compliance With Llm Reviewing Policy:**

Affirmed.

**Key Questions For Authors:**

1. How many of the generated OpenGrep rules strict enforce arbitrary naming conventions (as seen in Figure 8) versus purely structural or dataflow constraints?

2. Why does the Precision metric rely on exact line intersections, and how does this metric account for benign formatting differences introduced by the models?

**Limitations:**

Yes

**Strengths And Weaknesses:**

Strengths
- The scale of the tasks evaluated is highly impressive. As demonstrated in Table 1, the median instance requires editing 73 files and over 1,500 lines of code. This pushes the boundaries of agentic context management well beyond existing single-file or localized benchmarks.
- The integration of repository test suites for functional validation alongside LLM-generated, deterministically verified OpenGrep rules provides a smart, scalable alternative to manual rule-writing or purely syntactic diff-checking.
- By isolating the evaluation into an Instructed Track and an Open Track, the paper successfully disentangles a model's ability to follow complex editing instructions from its capacity to autonomously infer architectural intent.

Weakness
- Table 1 shows that the average instruction length in the Open Track is a mere 38 characters. Expecting an agent to "Refactor routing system" and perfectly replicate the exact architectural choices of the original human developers is an unrealistic benchmark. Figure 8 explicitly illustrates this issue: the positive rule checks for a variable declared exactly as DnsRecordType. If an agent implements a functionally superior refactoring but names the type DNSRecord, it fails the rule. This reduces the Open Track to a guessing game of author intent rather than an evaluation of code quality improvement.
- With only 100 instances, the statistical confidence in differentiating agent performance is compromised. As seen in Table 3, the Alignment score for Sonnet 4.5 in Open Track Multiplan is 8.5% with a wide CI of [4.5, 12.4], rendering it statistically indistinguishable from its Plan mode performance (9.4% [5.3, 13.5]). Additionally, Figure 4 reveals severe language imbalances (e.g., only 3 instances for C/C++).
- The Precision metric ($PREC$) defined in Section 4.1 relies on the exact intersection of added/removed lines with lines covered by OpenGrep rules. Any minor formatting changes (e.g., line breaks, comment additions) introduced by the agent will heavily penalize the score. This likely explains why $PREC$ remains universally low (under 20%) in the Open Track, as visualized in the right panel of Figure 7, making it an unreliable proxy for evaluating "unrelated changes".
- Since the Open Track relies heavily on golden-commit alignment, it fails to measure objective code improvements. The paper completely ignores established maintainability metrics which are standard in software engineering literature for evaluating the utility of a refactoring.
- The paper notes in Section 4.3 that Sonnet 4.5's performance slightly drops in Oracle Multiplan mode compared to Plan mode. However, there is no deeper analysis to determine whether the failure stems from the model generating worse plans, or from the Oracle failing to select the best available plan.

---

> ### Author Rebuttal · Authors · 2026-03-31
>
> We thank the reviewer for their positive review of our work. Below, we address the remaining questions
>
> **Q1: Is the replication of human refactoring decisions in the Open Track setting realistic?**
>
>
> We agree with the reviewer that there are many possible reasonable refactorings after discovering a deficiency in the code base. However, all refactorings covered in CodeTaste are quite significant, often covering thousands of lines of code. This indicates a high priority of the applied refactorings. We therefore expect that models are able to recover them independently and achieve a large IFR and alignment score.
>
>
> In particular in the Multiplan setting, we additionally steer the models in the direction of the desired refactorings. Even there, IFR is quite low. Our analysis in our answer to Q3 of reviewer B7bG highlights that this is a shortcoming of the LLMs and not a mistake by the LLM-as-a-Judge: the models majorly do not propose plans that are similar to the human-preferred refactorings. Meanwhile, high scores in this category are not impossible. GPT-5.2 manages to score 25% on IFR.
>
>
> This does not imply that the refactorings expected in CodeTaste are the *only* conceivable refactorings for a given codebase and in the open setting of the Open Track, we agree that other refactorings may be validly applied additionally. We therefore will focus the Open Track analysis on the IFR and alignment scores only in the next revision of the paper, and remove the focus from the precision score.
>
>
> **Q2: Which conclusions in the Open Track are significant? Could you highlight this more?**
>
>
>
> We refer the reviewer to our answer to Q1 of YpcX.
>
>
> **Q3: Is there a strong effect due to language imbalances?**
>
>
> First of, we would like to highlight that our work considers a remarkably high variety of programming languages. As discussed in our Related Work section, our work covers 6 distinct languages, while prior work focuses only on singular languages. We thus expect an overall smaller effect of programming language choice than in other works, even if the amount of instances is not perfectly balanced.
>
>
> Nonetheless, we investigated the impact of programming languages on model performance. We conducted a breakdown of scores per language. Overall, we see no consistent or significant impact of language on performance. The detailed results can be found in our answer to Q3 of reviewer fyHi.
>
>
> **Q4: How sensitive is the PREC score to code formatting and comments?**
>
>
> We thank the reviewer for highlighting this important nuance when measuring code diffs. We concretely update our precision metric to explicitly not count lines that fall into one of the following four categories:
> - Whitespace
> - Comments
> - Documentation files
> - Configuration files
> - Vendored artifacts and build directories
>
> Overall, we observe an increase in precision scores of on average 4% in the Instructed Track and 1.5%-2% in the Open Tracks. Notably, Qwen3 increases its score by 5% only due to ignoring configuration files, and another 3% after ignoring comments and documentation.
>
>
> **Q5: Would traditional metrics be beneficial in the Open Track to complement the IFR and Pass rates?**
>
>
> We don’t think that adding additional, generic software engineering metrics add much to our evaluation. Our target is primarily to evaluate explicit alignment with human developer refactoring choices. With the help of our LLM-based construction pipeline, we are able to evaluate this at a large scale with highly individualized rules. This is in stark contrast to generic metrics, which are themselves highly debated in the literature [1,2], because they are not strictly based on concrete findings regarding ease of code writing or understanding. Additionally, refactorings may not even be described effectively by such metrics, for example the migration from a deprecated method would not be represented in cyclomatic complexity or similar metrics. We therefore argue that the alignment as measured by our constructed rules is a much stronger signal for the capability of LLMs at reconstructing human refactoring choices.
>
> [1] Shepperd, *A critique of cyclomatic complexity as a software metric*, 1988
> [2] Hao et. al, *Complementarity in software code complexity metrics*, 2026
>
>
> **Q6: What is the reason behind the Claude Sonnet 4.5 score decrease in Multiplan mode?**
>
>
> We refer the reviewer to our answer to Q4 of reviewer B7gB.

---

> > ### Author Rebuttal · Reviewer_wheP · 2026-04-03
> >
> > Thanks for your rebuttal.  I will keep my score.

---

### Official Review · Reviewer_YpcX · 2026-03-03

**Soundness:** 2
**Presentation:** 3
**Significance:** 3
**Originality:** 3
**Overall Recommendation:** 4
**Confidence:** 3

**Summary:**

This paper introduces CodeTaste, a benchmark of 100 real-world, multi-file refactoring tasks mined from 87 open-source repositories across 6 languages. Evaluation combines repository test suites (PASS) with rule-based instruction following rate (IFR) into a multiplicative alignment score (A). The benchmark features two tracks: an instructed track with detailed refactoring specifications, and an open track where agents must autonomously perform the refactoring task given only a vague prompt.

**Compliance With Llm Reviewing Policy:**

Affirmed.

**Final Justification:**

The author fully resolve my concerns, and the promise of additional experiments is convince.

**Key Questions For Authors:**

1. With only 100 instances, the confidence intervals for key metrics are quite wide. Have you conducted any  analysis or experiments to determine whether this sample size is sufficient to reliably distinguish model performance(e.g., std of evaluation results), or had a plan to scale the benchmark?

2. The current evaluation is limited to frontier models. Could you provide more results on different-size models?

3. Have you performed any human audit or systematic analysis to get the coverage of rules? Is the uncovered portion dominated by certain categories? Or the rules basically cover all the  refactoring instances? Understanding this gap is important for interpreting whether low model scores reflect genuine misalignment or simply evaluation blind spots.

**Limitations:**

yes

**Strengths And Weaknesses:**

## Strengths

**Soundness:**
- The benchmark construction pipeline is rigorous and well-engineered. The three-stage data curation process  narrows down millions of commits to 100 high-quality refactoring instances. The rule discovery mechanism is particularly noteworthy: an LLM agent iteratively proposes OpenGrep rules with automated validation against both the base and golden codebases, producing a substantial number of rules.
- The evaluation metrics are carefully designed to capture multiple dimensions of refactoring quality. PASS gates on functional correctness, IFR (including IFR+ and IFR-) measures correspondence to refactoring intent, and PREC quantifies how much of the patch is relevant to the intended transformation. Together, these metrics provide a fine-grained multidimensional comparison between human refactorings and model-generated code.


## Weaknesses

**Soundness:**
- The benchmark contains only 100 instances. While authors acknowledge the high cost of evaluation, this limited sample size results in instability, which weakens the statistical reliability of inter-model comparisons. It remains unclear whether the observed performance differences are robust or could shift  with a larger test set.

- The model coverage is limited to a few frontier-scale models (e.g.,GPT-5.2). Including additional medium-scale and small-scale models would provide a more complete picture of how refactoring capability scales with model size and capability.

-  Rules are entirely generated by LLM agents, and while the registration mechanism enforces basic validity constraints, there is no systematic evaluation of rule quality beyond this. An analysis of rule quality would strengthen confidence in the evaluation framework.

---

> ### Author Rebuttal · Authors · 2026-03-31
>
> We thank the reviewer for their review and highlighting the quality and metrics of our benchmark. Below, we address the raised questions by the reviewer.
>
>
> **Q1: Can you please highlight which significant conclusions can be drawn from the results?**
>
>
> We thank the reviewer for raising this important point. To enable direct estimation of significance for all results in the paper, we have provided confidence intervals in all figures in the submitted paper. Below, we provide results from an additional pairwise analysis.
>
>
> Concretely, we conducted a pairwise statistical analysis using a two-sided paired sign-flip permutation test with 100,000 permutations and used Benjamini-Hochberg FDR correction with a significance threshold q < 0.05.
>
>
> In particular, we highlight the following 3 results.
> 1. In Instructed Track and Open Track Multiplan, GPT-5.2's lead over all other models is highly significant ($p = 10^{-5}$, and $p = 2 \times 10^{-5}$ respectively).
> 2. In Open Track Multiplan, Sonnet 4.5 is significantly better than the remaining models, except for MiniMax M2.7.
> 3. In both tracks, Qwen3 is significantly outperformed by all other models.
>
>
> We will add the detailed results in our next revision of the paper.
>
> **Q2: Could you provide more results on a medium-sized model?**
>
>
> To complement our analysis of frontier models and the Qwen 30B model, we ran Minimax M2.7, which has 229B parameters. We use the recommended Claude Code harness. Across all settings, it has a similar alignment score and precision to Claude Sonnet 4.5 and GPT-5.1 Codex Mini, and thus significantly outperforms Qwen 30B. For example, in the instruct mode, it achieves an alignment score of 32% (33%, 35%, for Sonnet and GPT-5.1 Mini, respectively) and a precision of 63% (58%, 57%, for Sonnet, GPT-5.1 Mini, respectively). The detailed results can be found in the attached PDF in Table 1.
>
>
> **Q3: Please provide a detailed analysis of the quality of the rules.**
>
>
> In our latest revision of the benchmark, we enforce that rules match distinct patterns. In addition, we improved the resilience to syntactical variations via more systematic use of symbolic propagation and taint analysis. This improvement has resulted in a slight increase in the observed scores, but did not affect the observed trends.
>
>
> Our main quality concern is requiring guessing arbitrary conventions in additive rules. After the above revision, two thirds of the rules used by CodeTaste are reductive rules, and only the remaining third correspond to additive rules. Thus, as long as the model discovers and modifies the undesired coding patterns, it can achieve a score of over 66%. Most models do not achieve this score, even when including additive rules.
>
>
> We further perform a human analysis of the additive static analysis rules. The goal is to flag rules that enforce arbitrary naming conventions, thus forcing the model to guess correct solutions. We randomly select 25 additive rules across benchmark instances and provide them to a human expert and an LLM-as-a-judge (Gemini 3.1 Pro / 3 Pro / 3 Flash)  for review. The human expert and judges identify between 32% and 50% of the patterns as overly restrictive. This is in line with our estimates on lines 385-392. The agreement between the human and Gemini 3 Pro / Majority vote stands at 84%, with a Cohen's kappa coefficient of 0.675. The strong alignment between human and LLM-as-a-judge allows us to use the approach to remove the overly restrictive rules from our set of additive rules. We will incorporate this in the next revision of our paper. However, we do not expect a change in the overall trend of our results, as the upper bound of the improvements is around 4%.
>
>
> The full scores of our revised variant can be inspected in the attached table in https://anonymous.4open.science/r/r-1/.

---

> > ### Author Rebuttal · Reviewer_YpcX · 2026-04-01
> >
> > Thanks for your rebuttal. However, my question 1 is not fully resolved. I will keep my score.

---

> > > ### Author Response · Authors · 2026-04-03
> > >
> > > We thank the reviewer for their acknowledgement. We discuss the remaining concerns below.
> > >
> > > **AQ1: Will the results change with a larger test set?**
> > >
> > > To fully address the concerns of the reviewer, we plan to scale the benchmark to 200 instances and repeat our main experiments on this scaled dataset. Due to the significant cost and effort of evaluation, we will not be able to present additional results by the end of the discussion phase. However, we are confident that our core observations will not change, since the findings outlined in our answer to Q1 are highly significant.

---

### Official Review · Reviewer_B7gB · 2026-03-11

**Soundness:** 3
**Presentation:** 3
**Significance:** 4
**Originality:** 3
**Overall Recommendation:** 6
**Confidence:** 3

**Summary:**

This paper studies the problem of addressing architectural debt in codebases through functionality-preserving refactoring. To evaluate whether LLM agents can perform refactorings aligned with human intent, the authors introduce a challenging benchmark with two evaluation tracks: an **instructed track**, where agents are given a detailed refactoring goal, and an **open track**, where agents must propose a human-aligned refactoring based only on a general focus area rather than precise instructions.
The dataset is constructed by mining pull requests from the GitHub Archive using keyword and popularity filters. The authors then apply additional filtering and LLM-based scoring to identify large, repository-wide, behavior-preserving refactorings. The resulting benchmark contains 100 instances spanning 87 repositories and 6 programming languages. On average, each instance requires edits to 73 files and approximately 1.5K lines of code. Functional correctness is validated using extensive test suites averaging nearly 1.6K tests, along with around 100 static checks for desired and undesired code patterns.
The authors evaluate four coding agents using the test suites to verify functionality preservation and **OpenGrep**-based static rules to assess alignment with desirable and undesirable refactoring patterns. Results show that frontier agents perform relatively well on the instructed track, but performance drops sharply on the open track, accompanied by a decline in instruction-following precision.
The paper also explores alternative interaction settings, including a **planning (propose-then-implement)** mode and an **oracle multi-plan** setting in which agents generate multiple candidate plans and an oracle judge, with access to the ground-truth refactoring, selects the most aligned plan. Although these settings yield notable relative improvements, overall performance remains low, highlighting the challenging nature of the benchmark.

**Compliance With Llm Reviewing Policy:**

Affirmed.

**Final Justification:**

I believe this is a really strong paper for code refactoring, and despite the small size of the released benchmark, the shared pipeline is reproducible from the details in the appendix and could be used to collect more data. Compared to existing refactoring benchmarks out right now for code-refactoring, it is relatively unique as it measures both refactoring quality and functionality preservation in general contexts. While benchmarks like SWE-Perf or SWE fficiency measure the refactoring quality through performance speedups, the scope is a lot more specific and compared to RefactorBench and SWE PolyBench, their dataset is more complex and covers more languages, making it a significant contribution along with the data collection pipeline, which could be used for training as well.

**Key Questions For Authors:**

1. Is it possible to get an estimate of the number of restrictive static rules, like mentions of code elements, that should appear by using an LLM judge-based eval?
2. **Clarification:** How do you go from the 640 candidates mentioned in Appendix A.1 to the final 100 instances?
3. How good/reliable is the Oracle Judge for the Oracle Multiplan setting? Why is Claude 4.5 Sonnet bad in this setting when it itself is the Oracle Judge? Can you do a comparison between it and human judgment for a few instances?

**Limitations:**

While limitations are covered in various sections of the paper there is no dedicated limitations section and adding one would be good.

**Strengths And Weaknesses:**

### Strengths:

1. **Challenging benchmark design.** The paper proposes a challenging benchmark with two evaluation tracks for assessing code refactoring capabilities of agents: an easier instructed setting with detailed refactoring instructions and a more open-ended setting where agents are given only high-level focus areas.
2. **Well-defined evaluation metrics.** The paper introduces metrics to evaluate both functional correctness preservation and alignment with human refactoring patterns. In particular, it uses a fine-grained instruction-following rate (IFR) metric based on static code checks for both desired and undesired refactoring patterns.
3. **Practical data construction pipeline.** The authors develop a useful pipeline for identifying large-scale, functionality-preserving refactorings from pull requests, generating refactoring instructions, constructing static rules to evaluate refactoring quality, building execution environments, and generating tests to verify functional correctness.
4. **Comprehensive empirical evaluation.** The paper evaluates several popular coding agents across both tracks and additionally studies planning-based settings, including propose-then-implement and picking the oracle plan (giving the coding agent the best shot) for the challenging open track. It also includes qualitative analysis of failure modes in the open setting.

### Weaknesses:

1. **Potentially restrictive static checks.** Some of the low precision observed in the open track, as reflected by the IFR metric, may partly stem from overly restrictive static checks, for example rules that expect specific element names in the code.
2. **Large difficulty gap between tracks.** The instructed and open tracks differ substantially in difficulty, resulting in a large performance gap. Introducing an intermediate setting with partially specified instructions, more detailed than high-level focus areas but less prescriptive than the instructed track, could provide a more realistic evaluation of agent capabilities.
3. **Oracle selection in the multi-plan setting.** In the oracle multi-plan setting, the oracle judge may not always select the most optimal plan among the proposed candidates, which could limit the reliability of the reported upper-bound performance.

---

> ### Author Rebuttal · Authors · 2026-03-31
>
> We thank the reviewer for their overall positive review of our work. Below, we address the remaining questions and provide results of additional investigations.
>
>
>
> **Q1: Can you provide an evaluation of how frequently the rules are overly restrictive?**
>
>
> We agree with the reviewer that a systematic analysis of the rules is necessary to assess and ensure the quality of the rules. We refer them to our answer to Q3 of reviewer YpcX for our detailed analysis.
>
>
> **Q2: Would it make sense to add an intermediate track between Instructed and Open?**
>
>
> We thank the reviewer for this interesting suggestion. However, we argue that the current two track modes conceptually describe two very clear metrics: The Instructed Track measures whether agents can execute a specifically detailed refactoring, even if it is extensive. The Open Track on the other hand measures the ability to both discover and execute a refactoring. While Instructed Tack focuses on helpfulness as a tool, the Open Track is designed to assess full autonomy of coding agents. We don’t think that a middle ground, where human supervision is still present and only provides loose guidance, is very realistic and able to add much insight in this setting.
>
>
> **Q3: How reliable is the Oracle Judge for the Oracle Multiplan setting?**
>
>
> We conduct an experiment to measure how reliable the LLM Oracle judge is, by assessing whether it reliably picks plans that result in higher $\text{IFR}$.
>
>
> We consider all GPT-5.2 and Sonnet 4.5 model-instance pairs where the observed execution outcome differs meaningfully between plan mode $P$ and multiplan mode $M$, i.e., $|\text{IFR}\_{\text{M}} - \text{IFR}\_{\text{P}}| > 0.05$. This yields 49 instances for GPT-5.2 and 40 for Sonnet 4.5. Let $p_{P}$ denote the plan produced in single plan mode, and let $p_{M}$ the multiplan candidate that was selected and executed in the multiplan run. Let $p_{W}$ denote a plan that produced a higher $\text{IFR}$ score. We then run a six-plan judge evaluation in which the judge sees the five multiplan candidates together with $p_{P}$, presented in randomized order.
>
>
> Let $p_J$ denote the candidate selected by the judge. We distinguish three outcomes: $p\_J = p\_{P}$, $p\_J = p\_{M}$, and $p\_J \notin \{p\_P, p\_M\}$, where the last case means that the judge selects one of the other four multiplan candidates. Across all instances, we find that the judge reliably picks the best plan, with $P[p\_J = p\_W] = 72\\%$. Note that in many instances all plans are not well aligned with the original refactoring intent, in which case the judge may randomly pick a non-winning plan without real impact on $\text{IFR}$. We manually review 20 of the judge's reasoning traces and find well-justified differentiations between plans where applicable.
>
>
> **Q4: Why does Claude Sonnet lose performance in Multiplan?**
>
>
> We conduct a second experiment where we again pass all six plans to the judge and ask how often the judge prefers $p_P$ over $p_M$ for GPT-5.2 and Sonnet 4.5. We exclude 32 degenerate cases where $\text{IFR}\_{\text{M}} = \text{IFR}\_{\text{P}} = 0$.
>
>
> We find that for GPT-5.2, the multiplan plan $p_M$ is preferred in 56% of cases, and the single plan $p_P$ in only 35% of cases (in 9% of cases plans other than $p_P$ or $p_M$ are picked). Meanwhile, for Sonnet 4.5, the direct plan is preferred in 63% of cases, while the multiplan is preferred in only 30% of cases. This indicates that the lower $\text{IFR}$ for Claude Sonnet 4.5 in Multiplan mode is indeed to on average less aligned plans in multiplan mode.
>
>
> **Q5: Please clarify how the pipeline reduces 640 candidate instances to the final 100.**
>
>
> The pipeline obtains 640 instance candidates, which we manually assess for the final selection of 100 instances. Concretely, our assessment consists of two steps: a) manually assessing the quality of the generated rules and b) confirming the correctness of the generated execution environment. We keep the first 100 instances that pass both of these checks to reduce the manual effort and final cost of the pipeline. We thank the reviewer for pointing out this missing detail, and will clarify it in the next revision of the paper.
>
>
> **Q6: Please add a dedicated "Limitations" section to the paper.**
>
>
> We thank the reviewer for pointing out this missing section. We will make sure to integrate it in the next revision of our paper, basing the content on the following points which are currently spread out across the paper:
> - The size of the dataset is limited to 100 instances, making smaller differences, such as between models in Open Track mode less significant.
> - The language distribution is uneven, which may skew the results of our evaluation.
> - The LLM-based generation of rules, execution environments and their use as quality filters may introduce systematic biases that are not yet accounted for.

---

> > ### Author Rebuttal · Reviewer_B7gB · 2026-04-04
> >
> > Most of my concerns were resolved. For 5 and 6, these details should definitely be added to the paper. For 4, I want a deeper understanding of why spending more compute to get potentially better plans overall worsens Claude 4.5 Sonnet's performance. Is the selection by the Oracle judge the source of this issue, or does Claude produce worse plans because you ask it to produce more plans? Overall, I choose to keep my positive assessment of this paper.

---

> > > ### Author Response · Authors · 2026-04-04
> > >
> > > We thank the reviewer for their acknowledgement. We discuss the remaining concerns below.
> > >
> > > **AQ1: Is the selection by the Oracle judge the source of this issue, or does Claude produce worse plans because you ask it to produce more plans?**
> > >
> > > The experiments described in the answers to Q3 and Q4 show that if the oracle judge from multiplan oracle mode is **additionally** presented with the single plan ($P\_P$) from Plan Mode alongside the 5 initial plans from multiplan mode (the selected plan $P\_{M}$ and the four other plans), it can reliably predict which plan ($P\_P$ vs. $P\_M$) will achieve a higher score. This suggests that the selection mechanism is not the bottleneck.
> > >
> > > Below, we simulate the obtained $\text{IFR}$, if the plan from single plan mode had additionally been part of the plans presented to the Oracle Judge in Multiplan Mode. The resulting IFR would have been higher than the IFR obtained in both Multiplan Mode and Plan Mode. This serves as additional evidence for the selection mechanism **not being the source of this issue**. Thus, we conclude that "Claude produces worse plans because we ask it to produce multiple plans".
> > >
> > > **6-Plan Simulation Details**
> > >
> > > Let $N=100$ be the total number of instances, and for each instance $i$:
> > > -   $P\_{P,i}$ is the plan from Plan Mode.
> > > -   $P\_{M,i}$ is the best-aligned plan from Multiplan Mode.
> > > -   $\text{IFR}(P\_{P,i})$ is the instruction following rate of instance $i$ in Open Track Plan Mode and $\text{IFR}\_{P\_P}= \frac{1}{N} \sum\_{i=1}^{N} \text{IFR}(P\_{P,i})$ is the reported Plan Mode average.
> > > -   $\text{IFR}(P\_{M,i})$ is the instruction following rate of instance i in Open Track Multiplan Mode, similarly $\text{IFR}\_{P\_{M}}= \frac{1}{N} \sum\_{i=1}^{N} \text{IFR}(P\_{M,i})$ is the reported Multiplan Mode average. Note that $P\_M$ is the choice made by the oracle judge from the **initial** set of 5 plans.
> > > -   $C\_i$ is the choice made by the oracle judge from the set of **6 plans** $\\{P\_{P,i}, P\_{M,i}, P\_{others,i}\\}$
> > >
> > > Now, we compute the $\text{IFR}$ on the 6-plan mode as follows: $\text{IFR}\_{\text{6-plan}} = \frac{1}{N} \sum\_{i=1}^{N} \text{IFR}\_{{\text{6-plan}},i}$, where
> > > $$\text{IFR}\_{{\text{6-plan}},i}= \begin{cases} \text{IFR}(P\_{P,i}) & \text{if } C\_i = P\_{P,i} \\\\ \text{IFR}(P\_{M,i}) & \text{if } C\_i = P\_{M,i} \\\\ \frac{\text{IFR}(P\_{P,i}) + \text{IFR}(P\_{M,i})}{2} & \text{otherwise} \end{cases}$$
> > >
> > > We find that the 6-plan mode with oracle judgement outperforms both the multiplan mode and plan mode (shown in the table below). This indicates that the problem of Sonnet performance in Multiplan mode is indeed the quality of proposed plans in multiplan mode.
> > >
> > >
> > > |            | $IFR\_{P\_{P}}$ | $IFR\_{P\_{M}}$ | $IFR\_{\text{6-plan}}$ |
> > > | ---------- | ------------- | ------------- | --------------------- |
> > > | GPT-5.2    | 15.5\%        | 24.3\%        | 24.9\%                |
> > > | Sonnet 4.5 | 20.2\%        | 16.8\%        | 20.9\%                |

---

### Official Review · Reviewer_fYhi · 2026-03-12

**Soundness:** 2
**Presentation:** 2
**Significance:** 2
**Originality:** 3
**Overall Recommendation:** 2
**Confidence:** 3

**Summary:**

This paper studies the ability to refactor large codebases of LLMs. Two modes are considered: the Instructed Track and Open Track. The authors constructed a pipeline to generate test instances from Github commits. Existing coding agents perform adequately on the Instructed Track but struggle on the Open Track, showing a gap in ambiguous settings. Models benefit from planning on the Open Track before editing a codebase.

**Compliance With Llm Reviewing Policy:**

Affirmed.

**Final Justification:**

After the rebuttal, I maintain my score and think the submission should be rejected. I still have doubts with how the refactoring evaluation system was set up. Therefore, I couldn't support the acceptance of the current submission.

**Key Questions For Authors:**

1. Can the authors explain what “Rules” are and how they are generated and verified?

2. Is there a breakdown for different programming languages? Do models exhibit very different success rates depending on programming languages?

**Limitations:**

yes

**Strengths And Weaknesses:**

Strengths

Code refactoring is an effective way to reduce technical debt of a codebase. Studying the automation of this important feature holds promise in improving the maintainability of a codebase. The fact that models perform reasonably well on the Instructed Track points to some level of practicality.

Weaknesses
1. While code refactoring is an important aspect of maintaining a software project, it is inherently ill-defined. There are best practices but they tend to be abstract which makes code refactoring hard to evaluate. This paper makes an effort to define metrics to capture adherence to golden patches. However, since golden patches are not unique (there are other valid refactorings), these metrics become unreliable. Sadly I don’t have better suggestions on how to evaluate this.

2. Some parts of the writing are unclear. The most important one is on the definition of “Rules”. I did not understand what rules are or how they are generated (Section 3.3). This poses great difficulty in understanding some subsequent metrics such as IFRs. I would appreciate it if the authors can provide more intuitive explanations for rules.

---

> ### Author Rebuttal · Authors · 2026-03-31
>
> We thank the reviewer for their review and highlighting how practically relevant our work is. Below, we address the raised questions by the reviewer.
>
>
> **Q1: Can you clarify how the correctness evaluation based on static analysis rules works?**
>
>
> We thank the reviewer for the comment. We provide a detailed explanation below, and will update the next revision of the paper to incorporate it. First, we would like to highlight that we use static analysis rules specifically to avoid comparison to specific golden patches. As the reviewer pointed out, reproducing a specific golden patch is overly specific and would not result in a useful benchmarking signal.
>
>
> OpenGrep is a fairly popular open source static analyzer which is based on SemGrep. Intuitively, static analysis rules represent abstract patterns in code. These patterns are similar to regular expressions for text, but applied on abstract syntax trees and dataflow rather than text.
>
>
> Concretely for OpenGrep, its queries resemble the target language, for example `a = …` matches any assignment to variable `a`. The rules also support matching groups such as `foo($A, $A)` which means a call to `foo` with the same parameter twice. The language matches not only syntactic patterns, but also considers data flow. For example the rule `foo(bar())` would match the sequence `x=bar(); foo(x)` since it is semantically equivalent. We leverage this property to produce general rules that accept variants of the same refactoring. Figure 8 has an example of such a rule.
>
>
> We create two classes of these rules: 1) reductive rules that describe undesired patterns, which should be removed from the code (e.g., use of deprecated methods), and 2) additive rules that describe desirable code patterns (e.g., use of modern APIs). As part of the benchmark construction, we ensure that rules about undesirable code patterns match removed code and rules about desirable patterns match the added code.
>
>
> As a concrete example, we also highlight a reductive and an additive rule in Figure 1 of the attached PDF at https://anonymous.4open.science/r/r-1/. The reductive rule matches usage of a deprecated, generic state hook in a TypeScript framework. The additive rule matches usage of an appropriate replacement hook. The agents in the example (GPT-5.2 and Sonnet 4.5) correctly replace one usage with the other, such that the removed lines match the reductive rule, and the added lines match the additive rule.
>
>
> **Q2: How reliable are the constructed rules for model evaluation?**
>
> To assess the reliability of the constructed rules, we would like to highlight that only additive rules can be unreliable. The reason is that reductive rules are confirmed during construction to match code that was changed during the refactoring. Thus any refactoring choice must remove the undesired code pattern. Meanwhile, additive rules could be overly restrictive, enforcing arbitrary rules. For our analysis on this topic, we refer to our answer to Q1 of reviewer B7gB.
>
>
> **Q3: Is there any effect of language on the refactoring results?**
>
>
> Below, we present a per language breakdown of the average alignment score over all models. Overall, we see no significant impact of language on performance. We present the full results in the table below. Note that the benchmark contains only 3 C samples, resulting in overall larger uncertainty. We will add the respective figure and our findings in the next revision of this paper.
> | Mode | C | Go | Java | JavaScript | Python | Rust |
> |---|---|---|---|---|---|---|
> | Instructed | 59.50% ± 24.53% | 34.97% ± 6.68% | 35.47% ± 10.00% | 35.91% ± 7.09% | 30.75% ± 9.45% | 44.96% ± 15.34% |
> | Open + Direct | 29.70% ± 23.98% | 6.11% ± 2.74% | 3.79% ± 2.68% | 5.62% ± 2.84% | 5.00% ± 4.27% | 1.51% ± 2.23% |
> | Open + Plan | 29.05% ± 22.80% | 7.09% ± 2.68% | 1.68% ± 1.94% | 9.23% ± 3.60% | 8.11% ± 5.50% | 9.63% ± 8.09% |
> | Open + Multiplan | 18.88% ± 20.76% | 7.79% ± 3.02% | 5.90% ± 3.72% | 9.24% ± 3.69% | 8.55% ± 5.42% | 8.70% ± 6.48% |

---

> > ### Author Rebuttal · Reviewer_fYhi · 2026-04-01
> >
> > >As part of the benchmark construction, we ensure that rules about undesirable code patterns match removed code and rules about desirable patterns match the added code.
> >
> > Were those rules automatically constructed from code diffs or did human craft those rules? "Code patterns" are ambiguous and deserve a precise definition.

---

> > > ### Author Response · Authors · 2026-04-03
> > >
> > > We thank the reviewer for their acknowledgement of our rebuttal and clarify the follow-up questions below.
> > >
> > >
> > > **AQ1: How were the static analysis rules created?**
> > >
> > >
> > > As detailed in Section 3.3 and Appendix A.2, the rules are generated by an autonomous LLM-based agent. Crucially, to avoid overfitting, the agent is *only* provided the text-based task description and the pre-PR state of the repository. It *never* has access to the golden commit or the refactoring code diff. The core idea is that the agent can propose rules, and our creation pipeline validates them against the base commit and the golden commit. The pipeline ensures that reductive rules only match on the base commit, and that additive rules only match on the codebase after applying the golden commit. This way, we obtain general rules that capture the intended refactoring without direct information flow from the golden commit into the rules.
> > >
> > >
> > > **AQ2: Can you provide a precise definition of code patterns?**
> > >
> > >
> > > Static analysis tools like OpenGrep are highly sophisticated machinery that combines a diverse set of static analysis techniques such as pattern matching and data flow analysis [1,2]. A formalization of its rule/pattern definition is beyond the scope of our work. We thus align with other works in the literature [3,4] to simply highlight the general representation of patterns using OpenGrep and illustrate one example in [the attached figure](https://anonymous.4open.science/r/r-1/rule_example.pdf).
> > >
> > >
> > > Note that for the majority of our work, we do not rely on specific details of how code patterns and static analysis engines work. We require only two properties: (i) that they are expressed in a short (domain-specific) language, (ii) that they allow application to source code, which produces a set of lines that they match.
> > >
> > >
> > > We will adapt our explanations in the next revision of the paper.
> > >
> > >
> > > [1] Bull et. al., *Semantic grep: regular expressions + relational abstraction*
> > > [2] [SemGrep / OpenGrep Rule Definition Documentation](https://semgrep.dev/docs/writing-rules/rule-syntax#operators)
> > > [3] Bennet and Counsel, *Semgrep\*: Improving the Limited Performance of Static Application Security Testing (SAST) Tools*
> > > [4] Tsigkourakos and Patsakis, *QRS: A Rule-Synthesizing Neuro-Symbolic Triad for Autonomous Vulnerability Discovery*

---

### Decision · Program_Chairs · 2026-04-30

**Decision:**

Accept (regular)

**Comment:**

The manuscript studies the problem of addressing architectural debt, complexity, code duplication, and more generally, code maintainability in codebases through functionality-preserving refactoring. The authors introduce a challenging benchmark to evaluate code refactorings under two settings: 1) an instructed track, where agents are given a detailed refactoring goal, and 2) an open track, where agents must propose a human-aligned refactoring based only on a general focus area rather than precise instructions. The dataset (of 100 instances) is derived from real pull requests from the GitHub Archive using a combination of keyword filters and LLM-based scoring, to identify large, repository-wide, behavior-preserving refactorings.

The authors present detailed evaluation results that show that frontier agents perform relatively well under the instructed setting, but performance drops sharply in the open setting, and in particular, declines in instruction-following precision. The paper also provides substantial analysis and additional results that yield insights into assistive and failure modes.

Overall, I like the contribution of this work, that tackles an important problem (and fills an important gap) in AI for software engineering. The reviewers were divided, but there is enough evidence that the reviewers appreciate the merits of the work. During the discussion phase, there was clear excitement in support of the paper. So, I recommed accept.